# Tropopause-level planetary wave source and its role in two-way troposphere-stratosphere coupling

Lina Boljka[1] and Thomas Birner[2]

[1]Department of Atmospheric Science, Colorado State University, Fort Collins, Colorado, USA
[2]Meteorological Institute, Ludwig-Maximilians-Universität München, Munich, Germany

**Correspondence:** Lina Boljka (lina.boljka@colostate.edu)

**Abstract.**

Atmospheric planetary waves play a fundamental role in driving stratospheric dynamics, including sudden stratospheric warming (SSW) events. It is well established that the bulk of the planetary wave activity originates near the surface. However, recent studies have pointed to a planetary wave source near the tropopause that may play an important role in the development of SSWs. Here we analyse the dynamical origin of this wave source and its impact on stratosphere-troposphere coupling, using an idealised model and a quasi-reanalysis. It is shown that the tropopause-level planetary wave source is associated with nonlinear wave-wave interactions, but it can also manifest as an apparent wave source due to transient wave decay. The resulting planetary waves may then propagate deep into the stratosphere, where they dissipate and may help to force SSWs. We find that when an SSW is preceded by the tropopause wave source, it is followed by a downward impact, decelerating the tropospheric zonal flow between 40-60°N several weeks later. A different downward impact is found following the SSWs preceded by surface wave source events, i.e., a deceleration of the tropospheric zonal flow polewards of 60°N several weeks later. This suggests that the tropopause wave source (along with the surface wave source) could potentially be used as one of the predictors of not only SSWs, but their downward impact as well.

## 1   Introduction

Planetary waves of the extratropical atmosphere are the largest Rossby waves spanning the size of the Earth's radius and longer, and are generally slowly varying (e.g., Burger, 1958; Phillips, 1963). It has been well established in the literature that planetary waves in the Northern Hemisphere (NH) have a source predominantly in the lower troposphere (hereafter surface wave source), primarily associated with orography, land-sea contrasts (e.g., Charney and Eliassen, 1949; Smagorinsky, 1953; Held, 1983; Held and Hoskins, 1985) as well as baroclinic instability (e.g., Charney, 1947; Eady, 1949; Hartmann, 1979) and nonlinear wave-wave interactions (e.g., Scinocca and Haynes, 1998; Domeisen and Plumb, 2012). These planetary waves can propagate upwards reaching the stratosphere, although the bulk of planetary wave activity is dissipated in the upper troposphere (e.g.,

Edmon et al., 1980). The remaining planetary wave activity may propagate deeply into the winter stratosphere where they can disturb the polar vortex via wave-mean flow interactions, decelerating the stratospheric westerlies (e.g., Charney and Drazin, 1961; Matsuno, 1970, 1971; Holton and Mass, 1976; Limpasuvan et al., 2004). Under extreme situations this results in sudden stratospheric warming (SSW) events (e.g., Schoeberl, 1978; Butler et al., 2015) or sudden stratospheric deceleration (SSD) events (Birner and Albers, 2017). SSDs are abrupt decelerations of the stratospheric polar vortex, whereas SSWs require a reversal of the stratospheric flow from westerlies to easterlies.

Wave-mean flow interaction is important for the onset of a stratospheric event, however anomalous surface wave forcing prior to the stratospheric event is not a necessary condition for SSDs or SSWs to occur (e.g., Plumb, 1981; Scott and Polvani, 2006; Hitchcock and Haynes, 2016; Birner and Albers, 2017; de la Cámara et al., 2017, 2019). That is, climatological tropospheric wave forcing can be sufficient, depending on the stratospheric state prior to the event. Indeed, only about a third of the SSWs are associated with a strong tropospheric precursor (i.e., wave activity fluxes exceeding 2 standard deviations; Birner and Albers, 2017; Lindgren et al., 2018; de la Cámara et al., 2019; White et al., 2019). This leaves two thirds of the SSWs that are not directly related to a strong tropospheric precursor, suggesting that their origin depends more on stratospheric conditions. This means that the stratosphere exerts some control over the tropospheric waves that can enter it (e.g., Scott and Polvani, 2004; Hitchcock and Haynes, 2016). Indeed, Scott and Polvani (2006); Sjoberg and Birner (2014) have shown that even for steady tropospheric forcing SSWs still occur. One of the suggested mechanisms for the onset of the SSWs is self-tuning resonance (e.g., Plumb, 1981; Matthewman and Esler, 2011; Esler and Matthewman, 2011; Lindgren and Sheshadri, 2020), however it is still debated in the literature (e.g., Dunn-Sigouin and Shaw, 2020).

The above studies suggest that an extreme surface wave source is not necessary for exciting a stratospheric event, and, e.g., Birner and Albers (2017) have further argued that the dynamics of the lowermost stratosphere (just above the tropopause) and tropopause inversion layer could be important in exciting the stratospheric events. These studies have also pointed towards a potential planetary wave source just above the tropopause (hereafter tropopause wave source) that precedes the stratospheric events. Indeed, de la Cámara et al. (2019) found a wavenumber one wave source at the tropopause preceding displacement SSD events regardless of a surface wave source in a comprehensive model and a quasi-reanalysis, whereas for split SSD events this was less clear. This is in contrast with any SSDs/SSWs (regardless of splits or displacements) preceded by a surface wave source, which may exhibit a tropopause wave source following the SSD/SSW events. Recent studies have also identified a climatological tropopause wave source (Birner et al., 2013; Dwyer and O'Gorman, 2017) on the poleward side of the subtropical jet stream. Birner et al. (2013) have suggested that it is caused by upscale cascade from synoptic to planetary scale waves when there is a poleward flux of enstrophy (a dynamical mechanism). On the other hand, Dwyer and O'Gorman (2017) have suggested that latent heat release related to convective processes (diabatic effects) can cause a wave source at the tropopause even when planetary waves are not present. These results suggest a presence of a tropopause wave source, which could be associated with the stratospheric events, however its dynamics and impacts have not yet been fully explored.

The stratospheric events described above are important as they can have a downward impact on the troposphere several weeks after the event (e.g., Baldwin and Dunkerton, 2001; Thompson et al., 2002; Hitchcock and Simpson, 2014), which is typically associated with a negative index in the Northern Annular Mode (NAM) or the North Atlantic Oscillation (NAO),

continental cold air outbreaks, and an equatorward shift of the extratropical jet stream. Since the tropospheric signal occurs several weeks later, the stratospheric events also provide a source of predictability of the tropospheric weather regimes beyond the typical weather forecast horizon (e.g., Tripathi et al., 2015; Domeisen et al., 2020). Furthermore, planetary wave sources that precede an SSW/SSD event can further be used as precursors for predicting the strong disruption of the polar vortex and their later downward impact (e.g., White et al., 2019). This suggests that a better understanding of the planetary wave sources and planetary wave propagation is important for a better understanding and prediction of SSD/SSW events as well as their downward impact. Here we focus on the relatively less explored tropopause planetary wave source, its dynamical origins as well as its impacts on the atmospheric dynamics, especially around SSD and SSW events.

The paper is structured as follows. Section 2 provides theoretical hypotheses for the origin of the tropopause wave source, section 3 provides the methodology, section 4 tests the hypotheses from section 2, and section 5 addresses the impact of the tropopause wave source on two-way stratosphere-troposphere interaction, including a comparison with the surface wave source impacts. Conclusions are given in section 6.

## 2 What are potential mechanisms for a wave source at the tropopause?

Source of atmospheric waves is often defined using the Eliassen-Palm (EP) flux (Eliassen and Palm, 1961) divergence, which emerges in wave-mean flow interaction theories within the transformed Eulerian mean perspective (e.g., Andrews and McIntyre, 1976; Edmon et al., 1980). To demonstrate the relations between the eddies and the mean flow we use a quasi-geostrophic (QG) framework based on interactions between linear waves and the mean flow. Note that despite the limitations of the linear wave assumption such a framework has been successfully used in the past for studying stratospheric and tropospheric dynamics as well as their coupling (e.g. Plumb, 2013). This is arguably due to the fact that wave-mean flow coupling is fundamentally non-linear in this QG framework. The QG equations for the eddies and the mean flow are then (e.g., Andrews and McIntyre, 1976; Edmon et al., 1980)

$$\frac{\partial \mathcal{A}}{\partial t} + \nabla \cdot \mathbf{F} = \mathcal{D} \tag{1}$$

$$\frac{\partial [u]}{\partial t} - f[v]^* = \nabla \cdot \mathbf{F} + \mathcal{S} \tag{2}$$

where

$$\mathbf{F} = \left( -[u'v'] \cos \phi, \frac{f[v'\theta']}{\partial [\theta]/\partial p} \right), \tag{3}$$

$\nabla = (\partial/a\partial \sin \phi, \partial/\partial p)$, $\mathbf{F}$ is EP flux, $\mathcal{A} \propto [q'^2]$ is wave activity ($q$ is QG potential vorticity (PV), its square is enstrophy), $\mathcal{D}$ and $\mathcal{S}$ are source-sink terms, $\nabla \cdot \mathbf{F} = [v'q']$ is EP flux divergence (in QG theory it is equal to the meridional PV flux), $[v]^*$ is the meridional component of the residual meridional circulation, $u$ is zonal wind, $v$ is meridional wind, $\theta$ is potential temperature, $f$ is Coriolis parameter, $p$ is pressure, $\phi$ is latitude, $t$ is time, prime ($'$) denotes perturbation from zonal mean, and square brackets ([.]) denote a zonal mean.

EP flux divergence is present in the equation for the waves (Eq. (1)) as well as in the zonal momentum budget (Eq. (2)). The positive EP flux divergence (via Eq. (1)) represents a wave source ($\nabla \cdot \mathbf{F} > 0$, associated with $\partial \mathcal{A}/\partial t < 0$, i.e., waves leave the region), and the negative EP flux divergence represents a wave sink ($\nabla \cdot \mathbf{F} < 0$, associated with $\partial \mathcal{A}/\partial t > 0$, i.e., waves enter the region). At the same time EP flux divergence appears in Eq. (2), which means that EP flux divergence can affect both the zonal mean zonal wind changes (acceleration ($\partial [u]/\partial t > 0$) if $\nabla \cdot \mathbf{F} > 0$ or deceleration ($\partial [u]/\partial t < 0$) if $\nabla \cdot \mathbf{F} < 0$) or residual meridional circulation changes ($f[v]^* > 0$ if $\nabla \cdot \mathbf{F} < 0$ or $f[v]^* < 0$ if $\nabla \cdot \mathbf{F} > 0$). The changes to the mean flow depend on the depth of the $\nabla \cdot \mathbf{F}$ forcing (e.g., Haynes et al., 1991), i.e., for shallow forcing, such as the surface or tropopause wave source, the zonal mean zonal wind response is weak and the response of the residual circulation dominates, whereas the opposite is true for the deep forcing, such as in the mid-stratosphere.

This illustrates the importance of the EP flux divergence as a wave source and sink, further emphasising the understanding of the origin of the wave sources that occur in the atmosphere. As mentioned in the introduction, the wave source at the surface has been studied extensively and can be related to the topography, land-sea contrasts, baroclinic instability etc. (e.g., Charney and Eliassen, 1949; Smagorinsky, 1953; Hartmann, 1979; Scinocca and Haynes, 1998; Held et al., 2002; Garfinkel et al., 2020), however little is known about the wave source at the tropopause (e.g., de la Cámara et al., 2019), which could have a potential impact on the two-way stratosphere-troposphere interactions (more in section 5). Note that the effects of topography, baroclinic instability etc. could also directly affect the upper-tropospheric stationary waves, though their effects further up (e.g., at the tropopause) are likely small, and thus not considered further. In terms of the origin of the tropopause wave source, we explore two dynamical mechanisms: (i) wave decay, and (ii) upscale cascade.

## 2.1 Wave decay

Fig. 1 shows a schematic of a reversible wave growth (panel (a)) and decay (panel (b)) at, e.g., tropopause, which can result in an apparent wave source there (e.g., Hoskins, 1983b). As the wave grows (panel (a)) its meridional movements bring low PV air polewards ($q' < 0$, $v' > 0$) and high PV air equatorwards ($q' > 0$, $v' < 0$), resulting in an overall negative meridional PV flux (i.e., $[v'q'] < 0$) and a negative EP flux divergence (i.e., EP flux convergence; recall $\nabla \cdot \mathbf{F} = [v'q']$ in QG theory). Conversely, as the wave decays (panel (b)) its meridional movements bring low PV air equatorwards ($q' < 0$, $v' < 0$) and high PV air polewards ($q' > 0$, $v' > 0$), resulting in an overall positive meridional PV flux (i.e., $[v'q'] > 0$) and positive EP flux divergence. If there is no wave breaking or other effects (e.g., no combination with upscale cascade - see below), this process is reversible and thus an integration over time leaves no positive or negative EP flux divergence (i.e., summing panels (a) and (b) leaves $\nabla \cdot \mathbf{F} = [v'q'] = 0$). Therefore, even if the positive EP flux divergence exceeds a set threshold and appears as though there is a wave source, this is merely representing a wave decay, and thus we will refer to it as an apparent wave source.

The apparent wave source at the tropopause could also be caused by the waves entering the region, resulting in negative EP flux divergence ($\nabla \cdot \mathbf{F} < 0$), and later exiting the region, resulting in positive EP flux divergence ($\nabla \cdot \mathbf{F} > 0$). This again leads to $\nabla \cdot \mathbf{F} = 0$ when integrating over time. Note, however, that the exact causes of wave growth and decay are not a subject of this study (see, e.g., Hoskins, 1983b).

## 2.2 Upscale cascade via wave-wave interactions

Fig. 2 shows a schematic of upscale cascade (occurring via wave-wave interactions) in a vertical cross-section. Assume a wave source (positive EP flux divergence) in the lower-to-mid troposphere on the poleward side of the jet stream that can generate waves of various zonal wavenumbers (in the schematic waves of zonal wavenumbers 2, 4, 6 are considered, but in reality they are not limited to those wavenumbers) (see panel (a)). The waves can propagate upwards away from the wave source and dissipate at the tropopause (negative EP flux divergence, wave sink). However, as these waves break at the tropopause they may interact with each other nonlinearly, which can lead to upscale cascade.

In the schematic the upscale cascade occurs when the $k = 4$ and $k = 6$ waves interact (where $k$ is zonal wavenumber), leading to $k = 2$ wave generation, and with it a wave source in $k = 2$ occurs (positive EP flux divergence at the tropopause; see panel (b)). If the newly generated wave source for $k = 2$ waves is equal to the wave sink in $k = 4$ and $k = 6$ then the only mechanism at play is upscale cascade. However, if there is a pre-existing wave ($k = 2$) at the tropopause (either a wave with a source below the tropopause (e.g., a stationary wave; as shown in the schematic with an upward grey dashed wiggly arrow) or a growing/decaying wave from Fig. 1) it can interact with newly generated (via upscale cascade) $k = 2$ waves leading to resonance (via selective interference in triad interactions; see, e.g., Chapter 8.1.2 in Vallis, 2006) and amplification of the wave source at the tropopause (thus the wave source at the tropopause in panel (b) is stronger than the wave sink in panel (a)). Note that the resonance considered here occurs via triad interactions and is thus different from the self-tuning resonance[1] mentioned in the introduction. The triad interactions described here are meant to represent wave-wave interactions of a finite number of distinct large scale waves, rather than the turbulent interactions across a quasi-continuous range of wave numbers (as in 2-D turbulence theory). Interactions among waves of the same wavenumber is here considered to be a distinct process, not part of the triad interactions, which may give rise to resonance. While resonance is one possible mechanism for the wave source amplification, other diabatic and non-conservative effects may also lead to an amplification of the wave source at the tropopause (as denoted in Fig. 2b). As a wave at the tropopause is generated, it can propagate in any direction (as denoted in the schematic by $k = 2$ wave propagation), including upward, potentially disturbing the polar vortex in the stratosphere. Note that not all energy from small scale waves is lost in this process, thus some smaller scale waves can still be seen exiting the tropopause wave source/sink region (see $k = 6$ wave propagation in panel (b)). Furthermore, partial energy loss may occur during triad interactions due to dissipation. Nevertheless, this process results in a clear planetary scale wave source at the tropopause. Here note that while the wave source occurs at the tropopause, the synoptic/planetary scale waves responsible for its growth can originate in the troposphere, thus the troposphere (even if the waves are weak) can indirectly affect the stratospheric dynamics.

## 2.3 Other potential mechanisms

The wave decay and upscale cascade (with potential amplification via resonance or non-conservative and diabatic effects) are not the only possible mechanisms causing the wave source at the tropopause in the real atmosphere and are also not mutually

---

[1]Self-tuning resonance refers to the resonant interaction between a free travelling wave in the stratosphere with the forced wave propagating up from below (see Plumb, 1981, for further details).

exclusive. This means that while there can be upscale cascade and wave decay alone (as discussed above), they can also act together or they are accompanied by other processes, such as diabatic processes (e.g., latent heat release as noted in Dwyer and O'Gorman (2017) or cloud radiative effects - see, e.g., Albers et al. (2016), Fig. 14, which shows $k = 1$ (displacements) and $k = 2$ (splits) pattern in outgoing longwave radiation, potentially suggesting a role of cloud-radiative effects in planetary wave forcing); a meridional or vertical migration of waves before an upscale cascade or wave decay occurs (e.g., a wave source occurs polewards from the wave sink, thus resulting in net wave source in the poleward region; Birner et al., 2013); or other currently unknown dynamical processes. Since one of the main interests of this study is the dynamical origin of the tropopause wave source, we use a dry dynamical core model (described in section 3.1), which lacks the diabatic processes (such as latent heat release or cloud radiative effects), allowing us to assess the dynamical causes of the wave source only, and thus the diabatic effects are not discussed further in this study.

The two mechanisms (wave decay and upscale cascade) for the origin of a tropopause planetary wave source are tested in section 4, whereas section 5 investigates the impact of wave sources on the two-way stratosphere-troposphere interaction.

## 3  Methods

### 3.1  Model and data

The numerical model used for this study is the dry dynamical core version of the Geophysical Fluid Dynamics Laboratory (GFDL) model with a spectral dynamical core. The model configuration follows Held and Suarez (1994) with some modifications. The model is forced through Newtonian relaxation of the temperature field to a prescribed equilibrium profile, with linear frictional and thermal damping. We use a stratospheric perpetual solstice configuration, following Polvani and Kushner (2002)'s weak polar vortex forcing ($\gamma = 2$) with a troposphere-to-stratosphere transition at 200 hPa (as used in Sheshadri et al., 2015) and a zonal wavenumber 2 mountain with 2 km height following Gerber and Polvani (2009). The transition level (Sheshadri et al., 2015) is used as the tropopause layer is too deep in the Polvani and Kushner (2002) model configuration. Note that the tropospheric equilibrium temperature profile was not modified (i.e., follows Held and Suarez, 1994); only the stratospheric profile was. The model resolution is T63 (1.875-degree horizontal resolution at the Equator) with 50 varying vertical levels between 1000 hPa and 0 hPa (with the top half-level at $\sim 7$ Pa), and is run for 50005 days, of which the first 300 days are taken as a spin-up period. The zonal mean zonal wind climatology is shown in, e.g., Fig. 8a (black contours) below. The data are analysed as zonal mean and daily mean (from four-times-daily resolution - the eddy fluxes are first computed at 6-hourly resolution and then averaged over 24 h). This type of model is used here as it has been used in the past for mechanistic studies of coupled troposphere-stratosphere dynamics. It allows us to isolate large-scale dynamical processes important for the tropopause wave source and its impacts.

As one of the main interests of this study involves the dynamically driven transient behaviour at the tropopause (forced via, e.g., upscale cascade or wave decay mechanisms from section 2), the model configuration differs from the conventionally used strong polar vortex ($\gamma = 4$) with a tall mountain (i.e., 4 km) configuration (e.g., Sheshadri et al., 2015) for the following reasons. If the orographic forcing has a too strong amplitude, there appears to be a direct impact from stationary waves at the

185 tropopause (e.g., see Figs. 5c and 6a in Gerber, 2012), providing a stationary wave source near the tropopause, which leads to a relatively strong climatological positive EP flux divergence there. The goal here is to minimise such a direct impact near the tropopause. Therefore, we have weakened the surface planetary wave forcing, such that its forcing region is limited to 2 km height, which then required a weakening of the polar vortex to obtain a more realistic stratospheric variability and SSD/SSW events with potential downward impact. This yields a weak climatological EP flux divergence at the tropopause (similar in

magnitude to the observed values but opposite in sign; not shown), however the model still exhibits strong planetary (mostly $k = 2$) wave variability. Our model setup effectively precludes diabatic wave generation at the tropopause, but these diabatic effects cannot be excluded in the real atmosphere as in, e.g., ERA-20C described below.

The model results are compared to the ERA-20C quasi-reanalysis (Poli et al., 2016; Martineau et al., 2018), which is provided by the European Centre for Medium-Range Weather Forecasts (ECMWF). The analysis period is 1 January 1900

to 31 December 2010 and data are analysed only for November to March period (i.e. boreal cold season) of every year. The data are analysed on a 1.25° horizontal grid between 1000 hPa and 1 hPa (37 vertical levels). Daily anomalies were computed by subtracting a long-term trend in seasonal cycle (following de la Cámara et al., 2019). While ERA-20C is not a proper reanalysis dataset (constrained by surface observations only), it provides reasonable stratospheric variability as well as good statistics, which is especially important for studying stratosphere-troposphere interactions (e.g., Gerber and Martineau, 2018;

Hitchcock, 2019; de la Cámara et al., 2019). The underlying model of the ERA-20C also has a very good vertical resolution (91 vertical levels) both in the troposphere and in the stratosphere. Note that we have performed the analysis below also on JRA-55 reanalysis dataset (Kobayashi et al., 2015; Martineau et al., 2018), which yielded qualitatively similar results to ERA-20C, but the statistics were poor (due to small sample sizes), and are thus omitted (for brevity). Note also that while ERA-20C (constrained by surface observations only) yields qualitatively similar results to JRA-55, the results are less well-constrained

than for full-blown reanalyses (such as JRA-55).

### 3.2 Indices

SSDs are computed following Birner and Albers (2017); de la Cámara et al. (2019), by finding the largest 10-day drop of the 10 hPa zonal mean zonal wind ($[u]$) averaged between 45 and 75°N (i.e., $\Delta[u]/10\text{days}$), after exceeding the 2-standard deviations ($2\sigma$) threshold (in ERA-20C that is 20 m/s in 10 days, whereas in the model it is 13 m/s in 10 days). The index is then defined

as the mid-point of the deceleration, and the events have to be separated by at least 20 days. Fig. 3a shows a composite of the standardised stratospheric zonal mean zonal wind anomaly at 10 hPa averaged between 45 and 75°N in the model and in ERA-20C with the centre date during the largest deceleration (i.e., lag 0 is the SSD index), and averaged over all SSD events in each dataset. The evolution is overall similar, with the model deceleration events weaker on average than the ones in ERA-20C, and strengthening of the wind prior to SSD events (i.e., vortex preconditioning) is much weaker in the model ($[u] < 0.5\sigma$) than

it is in ERA-20C ($[u] > 1\sigma$; see section 5.2.1 for further discussion).

SSWs are defined as a subset of the identified SSDs as the first occurrence of the easterly zonal flow (at 10 hPa averaged between 45 and 75°N; cf., Butler et al., 2015) around the SSD event (examining between 5 days prior to the SSD and 20 days after).

The wave source events are defined using the Eliassen-Palm (EP) flux divergence (defined in section 2). Since one of the interests of this paper is the impact of the wave sources on the SSDs and SSWs in the stratosphere where only planetary waves are important, we compute the EP flux divergence that results from planetary scale waves (using Fourier transform). As we are interested in the impact on SSDs/SSWs and not in the distinction of split versus displacement SSW event, we focus on the dominant dynamics in each dataset. The dominant dynamics that is relevant in the stratosphere in ERA-20C is related to $k = 1$ planetary scale waves (e.g., de la Cámara et al., 2019), however when dry dynamical core models (using model configurations similar to Polvani and Kushner, 2002) are forced with $k = 1$ forcing (via topography), they do not exhibit SSWs (see, e.g., Table 1 in Sheshadri et al., 2015), thus the model is forced via $k = 2$ planetary waves, which then also represent the dominant stratospheric dynamics in the model. For these reasons, we then compute EP flux divergence for $k = 2$ waves in the model, whereas in ERA-20C we compute EP flux divergence for $k = 1$ waves.

To compute an index at the tropopause we first smooth the data with a 10-day running mean and then find the maximum positive 10-day mean EP flux divergence anomaly that exceeds $0.75\sigma$ threshold, and separate events by 20 days. Using stronger thresholds (e.g., $2\sigma$; as used in Birner and Albers, 2017) does not qualitatively change our final results (as it is the duration of the wave forcing that matters; Sjoberg and Birner, 2012), but does result in small sample sizes. The lower-stratospheric level close to the tropopause (in the following thus simply referred to as tropopause level), at which the index is computed, is chosen as the level at which EP flux divergence becomes positive and anomalously strong (exceeding the above threshold) and where it precedes the most SSDs: in the model that is at $\sim$200 hPa and in ERA-20C it is at $\sim$225 hPa. The level of the tropopause wave source is consistent with a slightly higher altitude of the extratropical tropopause in the model compared with ERA-20C. The latitudinal average of the wave source is over the latitudinal extent of the wave source region on the poleward side of the extratropical jet-stream (40-60°N in the model, and 45-75°N in the reanalyses). Recall that in the model all data are analysed, whereas in ERA-20C only the cold-season wave source events are identified.

As previous studies have used lower-tropospheric wave sources as precursors to SSDs, we use the same methodology as above for the lower tropospheric (surface) wave source, just that now we find the lower-tropospheric level at which EP flux divergence precedes the most SSDs: in the model that is at $\sim$685 hPa and in reanalyses it is at $\sim$700 hPa. Note that using vertical EP flux (from Eq. (3)) instead of EP flux divergence yields qualitatively similar results (in the sense that, e.g., zonal flow response is similar on average), however vertical EP flux is only a proxy for an actual wave source used here. Note also that using two different indices for the wave sources (i.e., at the tropopause and at the surface) allows us a comparison to previous studies as well as an analysis of the different atmospheric behaviour around, e.g., stratospheric events preceded by the tropopause and by the surface wave source events.

Fig. 3b,c show composites of the standardised EP flux divergence anomaly around wave source events (i.e., lag 0 is the wave source index) at $\sim$200 hPa and $\sim$700 hPa, respectively, for the model and ERA-20C. The figure shows that wave source events at the tropopause and at the surface are similar in standardised strength (peaking at $2\sigma$). These events also appear to be long-lived (i.e., exceeding the $0.75\sigma$ threshold for over 10 days) due to a 10-day smoothing applied before compositing over all events. Care must be taken in interpreting this apparent persistence, although there are many individual long-lived events

in the dataset (not shown). Note that the 10-day smoothing was applied as it is the 10-day mean wave forcing exceeding the threshold (e.g., $0.75\sigma$) that ultimately matters for the SSD generation.

The wave source indices described here are then used to test if SSDs are preceded by these wave sources, i.e., if SSD occurs within 12 days after wave source event then the SSD is preceded by the wave source event otherwise not. The 12-day horizon is similar to the 10-day horizon used in previous work (e.g., Birner and Albers, 2017; de la Cámara et al., 2019), which has been identified as a reasonable timescale for wave sources preceding an SSD. Here we again use 12-days to slightly increase the sample size, which does not qualitatively change the results. Some wave source events occur at the same time (i.e., surface

wave source occurs when tropopause wave source occurs, generally preceding it), which we have included in the analysis in section 4 (the origin of the tropopause wave source is largely unaffected by this distinction), but we have excluded it from the analysis of SSDs and SSWs in section 5.2 (they generally show a combination of impacts from both wave sources, obscuring their differences), leaving the results for the SSDs preceeded by the tropopause wave source only (i.e., not preceded by the surface wave source), and the SSDs preceded by the surface wave source only (i.e., no tropopause wave source following it

before the SSD occurs).

     The event statistics are provided in Table 1. We also show the number of events that have been used for each composite in panel-titles in figures below. We used a two-tailed t-test to perform a significance test, where non-significant (two-tailed p-value exceeds 0.05) values are shaded (colours) or excluded (arrows and contours). The data in figures are standardised (as in Fig. 3), i.e., normalised by standard deviation ($\sigma$), and the data were also smoothed with a 10-day running mean before plotting

(unsmoothed data yields similar, though noiser results).

### 3.3   Subjective analysis

In section 4 we test the hypotheses posed in section 2. While we can identify the wave source events and SSDs objectively (section 3.2), and the general picture that emerges follows the ideas presented in section 2, we subjectively identified a handful of cases in the model and in ERA-20C that clearly show the two mechanisms (upscale cascade and wave decay) around SSD

events. This is done as a way of showing that these mechanisms exist, since they can be obscured in an overall average or a randomly chosen case (more often than not they occur simultaneously).

     To identify upscale cascade only cases in the model, we compute $k = 2$ EP flux divergence and synoptic ($k \geq 4$) EP flux divergence at the level of the tropopause wave source. We then check the standardised anomalies of both EP flux divergences prior to an SSD event. If there is a negative synoptic EP flux divergence anomaly that is similar in amplitude (exceeding $0.75\sigma$

threshold) to the positive $k = 2$ EP flux divergence anomaly before (or at the same time as) the $k = 2$ EP flux divergence peaks, then it is classified as an upscale cascade. At the same time there must not be any strong negative $k = 2$ EP flux divergence preceding the positive $k = 2$ EP flux divergence peak. If there is no negative synoptic EP flux divergence or it is weak, this is not classified as upscale cascade. Note that in ERA-20C we test $k = 2, 3$ EP flux divergence instead of synoptic EP flux divergence, and we test $k = 1$ EP flux divergence instead of $k = 2$ EP flux divergence, since those are the leading contributions

to upscale cascade there.

<p style="text-align:center">9</p>

To identify the wave decay mechanism we perform similar analysis as for the upscale cascade mechanism, except that here the synoptic ($k = 2, 3$ in ERA-20C) EP flux divergence anomaly must not be strongly negative prior to the positive peak in $k = 2$ ($k = 1$ in ERA-20C) EP flux divergence anomaly. At the same time there must be a strong negative $k = 2$ ($k = 1$ in ERA-20C) EP flux divergence anomaly preceding the positive $k = 2$ ($k = 1$ in ERA-20C) EP flux divergence anomaly, which are similar in amplitude. This is then identified as the wave decay mechanism rather than the upscale cascade.

## 4   Evidence for the origin of the tropopause wave source

Section 2 suggested two possible mechanisms for the formation of the wave source at the tropopause: (i) wave decay (resulting in an apparent wave source), and (ii) upscale cascade. Here we test the two hypotheses using subjective analysis (section 3.3) in the model and in ERA-20C. Note that an average over all cases (objectively analysed as per section 3.2) shows indication of both mechanisms as well (see below), thus subjective analysis merely serves to highlight the mechanisms for clear cases of (i) and (ii).

Fig. 4 shows the model's $k = 2$ wave source/sink (anomalous positive/negative EP flux divergence; shading) and anomalous wave propagation (EP fluxes; arrows) in the top panels and synoptic ($k \geq 4$) wave source/sink in the bottom panels for lag-pressure composites over (a,b) all tropopause wave source events preceding SSDs (objective analysis), (c,d) subjectively selected events that demonstrate upscale cascade mechanism, and (e,f) subjectively selected events that demonstrate wave decay mechanism.

Fig. 4c demonstrates a wave source ($k = 2$) at $\sim$200 hPa around lag 0, i.e., the tropopause wave source. This wave source is not preceded by equally strong wave sink at the same level, excluding the possibility of strong wave decay mechanism (consistent with the definitions for subjective analysis; section 3.3). Moreover, Fig. 4d demonstrates a wave sink in the synoptic waves ($k \geq 4$) in the same region and at the same time as (or slightly before; seen in a case-by-case study; not shown) the wave source in Fig. 4c, suggesting an upscale cascade mechanism in generating the $k = 2$ wave source (consistent with the hypothesis discussed in section 2.2; see also schematic in Fig. 2). Note that the wave source ($k = 2$) is stronger than wave sink ($k \geq 4$), further suggesting a potential presence of other mechanisms such as resonance (as discussed in section 2.2). The EP fluxes (wave propagation; arrows) in Fig. 4c also demonstrate an amplification of wave propagation out of the wave source ($k = 2$) in equatorward (left tilt, i.e., not backwards in time) and upward direction, which can potentially disturb the stratospheric polar vortex (see section 5). Note that there is also a weaker surface $k = 2$ wave source present simultaneously with the tropopause wave source (Fig. 4c), which may reach the tropopause/lower stratosphere, where they can help amplifying the wave source (as discussed in sections 2.2, 2.3).

Fig. 4e demonstrates a wave decay mechanism, which results in an apparent wave source. As in Fig. 4c there is positive EP flux divergence (apparent wave source in $k = 2$ waves) at $\sim$200 hPa and lag 0, from which waves can propagate equatorwards and upwards, but the propagation is only significant on 90% significance level, thus not shown in Fig. 4e (95% significance level). Note that since this mechanism leads to an apparent wave source we do not necessarily expect wave propagation out of this wave source, unless it is upward wave propagation consistent with a wave entering the tropopause region from below

(wave sink) and then exiting it upwards (wave source). The apparent wave source ($k = 2$) here is weaker (compared with panel (c)) and preceded by a similar magnitude negative EP flux divergence ($k = 2$) at the same level, suggesting a wave decay mechanism (consistent with the definitions for subjective analysis; section 3.3), whereby the wave growth results in negative EP flux divergence and wave decay results in positive EP flux divergence (consistent with the hypothesis discussed in section 2.1; see also schematic in Fig. 1). Fig. 4f further demonstrates that upscale cascade does not occur in this case as the synoptic EP flux divergence shows a similar evolution to $k = 2$ EP flux divergence and not the opposite as in Fig. 4d, and it also does not pass the significance threshold.

Fig. 4a shows an average over all wave source ($k = 2$) events preceding SSDs and Fig. 4b shows the same but for the synoptic waves ($k \geq 4$). While the signal is weaker in an average over all events (considering different mechanisms involved in generating the wave source; discussed above; see also section 2) there is still an indication of upscale cascade (with a weaker wave sink in synoptic waves and a much stronger wave source in $k = 2$ waves at ~200 hPa and 0 lag), and there is a weak indication of wave decay as well, since there is a very weak negative EP flux divergence ($k = 2$) preceding the positive EP flux divergence ($k = 2$) at the tropopause. The signals of upscale cascade (via negative synoptic EP flux divergence) and wave decay (via negative $k = 2$ EP flux divergence preceding the wave source) are also likely obscured due to cancellations between different mechanisms. The $k = 2$ wave source is strong as both mechanisms lead to enhanced positive $k = 2$ EP flux divergence. Here note that the areas of negative ($k = 2$) EP flux divergence preceding the positive ($k = 2$) EP flux divergence in panel (a) as well as the negative synoptic EP flux divergence in panel (b) exhibit larger areas of significance if 90% significance level is used instead of the 95% (not shown).

Note that while we have selected only a few clear events to show that the two mechanisms truly exist, a case-by-case study revealed that the upscale cascade and wave decay are both common, and thus important for the wave source generation at the tropopause prior to SSDs. Also, the two mechanisms often occur simultaneously rather than separately, suggesting that more often than not both mechanisms play a role in generating a wave source at the tropopause (potentially amplifying each other).

Fig. 5 shows the same analysis as in Fig. 4 but for the ERA-20C data. In ERA-20C (as in the real atmosphere) the dominant stratospheric dynamics is related to $k = 1$ waves, thus the top row shows the results for $k = 1$ EP flux divergence and EP fluxes, whereas the bottom row shows EP flux divergence for $k = 2, 3$ waves, since these waves are more important in generating a $k = 1$ wave source at the tropopause than the synoptic waves in ERA-20C (objective analysis, similar to panels (a) and (b), for both the $k = 2, 3$ and the synoptic waves was performed to confirm this; not shown). As in Fig. 4c,d (for the model), Fig. 5c,d demonstrates upscale cascade with positive $k = 1$ EP flux divergence at ~200 hPa and lag 0 and negative $k = 2, 3$ EP flux divergence in the same region, but slightly before lag 0. Similarly, Fig. 5e,f demonstrates a wave decay mechanism for $k = 1$ waves (negative EP flux divergence precedes positive EP flux divergence, both having similar magnitude at ~200 hPa) in panel (e), with panel (f) confirming that upscale cascade is not taking place in this case (no clear wave sink in $k = 2, 3$ waves). As in Fig. 4a,b there is also an indication of upscale cascade and weak wave decay signal in an average over all tropopause wave source events preceding SSDs in Fig. 5a,b, however the wave propagation in ERA-20C is poleward (right tilt, i.e., not forward in time) instead of equatorward (partly because of different latitudinal averages in the model and ERA-20C; see also section 5.1). Here note that as in Fig. 4, the regions of significance increase if 90% significance level is used (not shown), which is true

especially for the panel (b), where there is a more significant negative $k = 2, 3$ EP flux divergence at the tropopause, as well as for the panel (e), where a more significant negative $k = 1$ EP flux divergence precedes the positive one at the the tropopause.

Unlike in the model, there is a weaker surface $k = 1$ wave source present in ERA-20C several days *before* the onset of the tropopause wave source (Fig. 5c), suggesting a possible wave propagation from the surface to the upper troposphere (a few days later, i.e., not simultaneous as in the model - Fig. 4c). Again, some of the waves may help to amplify the tropopause wave source. At positive lags there is a reappearance of the surface wave source in both datasets (though at different lags; Figs. 4c, 5c), and the negative EP flux divergence in the upper troposphere at short positive lags in ERA-20C (Fig. 5c) may be indicative of a sink or downward propagating waves (not seen in the model; Fig. 4c).

This section has thus shown that while different waves (in terms of their zonal wavenumber) cause upscale cascade and wave decay in the two datasets, both mechanisms are present in both datasets. As these mechanisms cause a wave source at the tropopause and there is upward wave propagation from this wave source (primarily following the upscale cascade mechanism), we now turn to its impact on two-way stratosphere-troposphere interaction.

## 5 Impacts of the planetary wave sources

### 5.1 Wave source events

Figs. 6 (model), 7 (ERA-20C) show lag-pressure composites over various event types for zonal flow anomalies (contours), EP flux divergence anomalies (wave sources/sinks; shading) and EP flux anomalies (meridional wave propagation, i.e., not in time; arrows). Panels (c) and (f) show the analysis for all tropopause wave source events and all surface wave source events, respectively.

Figs. 6c, 7c show a strong wave source (positive EP flux divergence anomaly) at the tropopause (lower stratosphere) with weak surface wave source occurring at the same time or slightly earlier. While there is some weak wave propagation out of the weak surface wave source, it largely decays in the upper troposphere and/or provides $k = 2$ (model) or $k = 1$ (ERA-20C) waves that can help amplifying the wave source there (see also sections 2.2, 4). Wave propagation (arrows) out of the tropopause wave source is largely upward, amplified and tilted (equatorward (left tilt) in the model; poleward (right tilt) in ERA-20C). The model and ERA-20C show different meridional direction in wave propagation, which is partly due to different meridional extents when averaging[2], and partly due to the wave source location. The latter means that if the wave source location is very close to strong merged subtropical-extratropical tropospheric jet stream (which acts as a wave guide) as in the model, the waves preferably propagate equatorwards in the selected latitudes. On the other hand, in ERA-20C the wave source is further poleward, in the proximity of the extratropical eddy-driven jet stream, and at the same time also closer to the stratospheric polar vortex above, thus poleward propagation of waves is more pronounced than in the model. There is also an indication of reduced upward wave propagation or in some cases even weak downward wave propagation (see downward EP flux anomaly)

---

[2]In the model we sample the equatorward propagating portion of the waves occurring within the 40-60°N latitudinal band; whereas in ERA-20C we largely sample the poleward propagation of the waves occurring in the 45-75°N latitudinal band; if we average data in the model over 45-75°N latitudinal band we recover more poleward wave propagation, though it is not as strong as in ERA-20C (not shown).

from the tropopause wave source in both datasets (clearer in ERA-20C). The zonal mean zonal wind anomalies differ between
the model and ERA-20C in the troposphere and in the stratosphere prior to wave source event (i.e., before lag 0), however after
lag 0 both datasets show wind deceleration in the upper stratosphere, suggesting that the tropopause wave source events may
impact the stratospheric winds.

Figs. 6f, 7f clearly show a strong wave source at the surface with weak tropopause wave source occurring slightly later
(consistent with the weak link between the surface and tropopause wave sources found in Figs. 6c, 7c). The wave propagation
from the surface wave source is largely upward only (less meridional tilting) and amplified as it leaves the wave source region,
reaching deep into the stratosphere. As in the case of the tropopause wave source events, there are similar differences between
the two datasets also for the surface wave source events, and both datasets suggest zonal flow deceleration in the stratosphere
(stronger than for the tropopause wave source events) following the surface wave source event (i.e., after lag 0), suggesting that
also the surface wave source can help decelerating winds in the stratosphere.

As both the tropopause and the surface wave sources have signals in the zonal flow in the stratosphere in both datasets,
we discuss below (section 5.2) the composites over the SSD and SSW events to further emphasise the impact of these wave
sources. Here note that only a small proportion of all wave source events precede SSDs (Table 1): ∼10% of the surface wave
source events result in an SSD in both datasets, whereas ∼10% (in ERA-20C) and ∼4% (in the model) of the tropopause wave
source events result in an SSD. The small proportion of wave source events that results in an SSD is likely a consequence of
the stratospheric dynamics, i.e., some preconditioning in the stratosphere is necessary for the wave source events to eventually
help decelerating the polar vortex (see, e.g., Scott and Polvani, 2004; Hitchcock and Haynes, 2016). Also, despite the small
percentage of wave source events that eventually lead to SSDs, they may be helpful for the understanding of the dynamics
around the SSDs. These events are hard to predict, and thus any new identified precursor may eventually help the understanding
and ultimately also the prediction of the SSDs.

## 5.2 Two-way stratosphere-troposphere coupling

Panels (a,b,d,e) in Figs. 6 (model), 7 (ERA-20C) show composites of EP flux divergence anomalies, EP flux anomalies, and
zonal mean zonal wind anomalies centred about an SSD event for (a) all SSD events regardless of wave source, (b) SSD events
preceded by tropopause wave source, (d) SSD events not preceded by any wave source, and (e) SSD events preceded by surface
wave source.

Figs. 6b,e, 7b,e are similar to Figs. 6c,f, 7c,f, in that they show similar location and amplitude of the tropopause and surface
wave sources as well as similar but much stronger wave propagation when composited over the SSD events. Since Figs. 6b,e,
7b,e are composited over SSD events, there is much stronger wind deceleration in the stratosphere than in Figs. 6c,f, 7c,f with
some suggestion of a downward impact (see below).

### 5.2.1 Polar vortex preconditioning

Prior to SSD events (i.e., negative lags in Figs. 6a,b,d,e, 7a,b,d,e) we can observe a positive stratospheric zonal mean zonal
wind anomaly, especially for SSDs preceded by surface wave source and SSDs not preceded by either wave source (Figs.

6d,e, 7d,e). This is generally referred to as stratospheric polar vortex preconditioning, which has also been used as one of the potential predictors of SSWs (e.g., Jucker and Reichler, 2018).

In ERA-20C there is a positive zonal mean zonal wind anomaly prior to SSDs (in Fig. 7a,b,d,e; see also Fig. 3a) with some notable (and significant) differences between the tropopause and surface wave source SSD events. Fig. 7b shows a gradual increase in the stratospheric zonal mean zonal wind, which is much weaker and occurring at lower levels (peaking at ∼20 hPa) than for the SSDs preceded by the surface wave source (Fig. 7e), where the enhancement of the stratospheric winds occurs more abruptly and at higher levels (peaking at ∼5 hPa).

A similar distinction can also be made in the model (Fig. 6b,e), where SSDs preceded by the tropopause wave source (Fig. 6b) lack a significantly positive stratospheric zonal mean zonal wind anomaly prior to an SSD (i.e., positive $[u]$ anomaly can still be present, but is not robust), whereas for the SSDs preceded by the surface wave source (Fig. 6e) there is a weak increase in the lower-stratospheric zonal mean zonal winds prior to an SSD (in contrast to much stronger increase in the mid-upper stratosphere in ERA-20C - Fig. 7e). Note that the difference between the two cases is much smaller (less significant) in the model than in ERA-20C. The lower-stratospheric increase in the zonal mean zonal wind (in Fig. 6a,d,e) is also consistent with Fig. 3a, where only weakly positive (and non-increasing) $[u]$ anomalies were found at 10 hPa, suggesting that in the model the stratospheric preconditioning might be more pronounced in the lower-stratosphere and thus less significant at higher levels (unlike in ERA-20C). While there are differences between the model and ERA-20C in terms of the strength and position of the positive zonal mean zonal wind anomaly in the stratosphere, they both suggest that SSDs preceded by the tropopause wave source require weaker stratospheric preconditioning than the SSDs preceded by the surface wave source. This distinction is likely a consequence of the tropopause wave source being present in the lower stratosphere already, thus additional preconditioning in that case is less important within the stratosphere (unlike for the surface wave source). This further suggests that strong polar vortex (strong negative PV gradients) combined with wave forcing, could be used together for predicting the SSDs/SSWs (see also Jucker and Reichler, 2018).

The preconditioning of the stratospheric polar vortex is not only important for SSDs preceded by the surface wave source, but also for SSDs that are not preceded by either of the wave sources (i.e., neither surface nor tropopause wave source; Figs. 6d, 7d), suggesting that stratospheric dynamics play an important role in the evolution of SSDs (e.g., self-tuning resonance; Plumb, 1981; Matthewman and Esler, 2011; Esler and Matthewman, 2011). Here note that in ERA-20C only $k = 1$ wave source events were examined, thus SSDs that are not preceded by $k = 1$ wave sources can still be preceded by $k = 2$ wave sources. Indeed, further analysis (not shown) confirmed that SSDs that are not preceded by $k = 1$ wave source events, show a significant positive $k = 2$ EP flux divergence at the tropopause (but not at the surface), suggesting importance of $k = 2$ waves (see also section 5.2.2). As only the $k = 2$ tropopause wave source (which occurs just above the tropopause in the lower stratosphere) is present, the above suggestion still holds, i.e., stratospheric internal dynamics likely matters.

While there is a positive stratospheric zonal mean zonal wind anomaly prior to an SSD, we can also observe positive tropospheric zonal mean zonal wind anomalies prior to SSDs for both wave sources and in both datasets (Figs. 6b,e, 7b,e),

consistent with linear theories of stratosphere-troposphere coupling, where tropospheric and stratospheric wind anomalies are equal in sign (when events are not explosive[3]).

### 5.2.2 Dynamical evolution around SSD events

As mentioned above, the SSD events can be split into different categories based on the precursory wave source events, such as (i) surface wave source, (ii) tropopause wave source, and (iii) no wave source. These wave sources tend to occur around the
455 beginning of zonal flow deceleration in the stratosphere and end as deceleration phases out (see negative EP flux divergence in the stratosphere, a qualitative proxy for $\partial[u]/\partial t$), and they last for over 10 days.

  When there is a tropopause wave source preceding an SSD (Figs. 6b, 7b), we can see wave propagation out of the wave source (at ∼200 hPa), especially in ERA-20C (Fig. 7b), whereas in the model there is weak wave propagation from the weak surface wave source present as well (which largely dissipates in mid-upper troposphere). Note that in ERA-20C there is only
460 a weak (non-robust) surface wave source present around the onset of the tropopause wave source (Fig. 7b), however in the model (Fig. 6b) the surface wave source is significant though weaker than for the surface wave source events (Fig. 6e). In the model (and also in ERA-20C, though not significant in Fig. 7b), the presence of tropospheric $k = 2$ ($k = 1$ in ERA-20C) waves can also contribute to potential resonance, occurring with upscale cascade at the tropopause (sections 2.2, 4), which further amplifies the wave source and subsequent wave propagation (note that wave source amplification is possible via non-
465 conservative or other effects as well). The waves that originate at the tropopause then dissipate in the stratosphere (negative EP flux divergence there), eventually leading to zonal flow deceleration.

  SSDs preceded by the surface wave source (Figs. 6e, 7e) show propagation of waves out of the surface wave source, which largely dissipate in the upper-troposphere/lower-stratosphere, especially in ERA-20C. While some of these waves likely make it deep into the stratosphere (indicated in the model, but less so in ERA-20C), there is wave amplification occurring within
470 the stratosphere as well, suggesting a role of stratospheric internal dynamics. In ERA-20C it is also possible that smaller scale waves occur via downscale cascade during the wave breaking (dissipation) process, thus no robust $k = 1$ waves are present in the lower-mid stratosphere. However, we found no robust smaller-scale waves (e.g., $k = 2, 3$) in that region, suggesting it may be case dependent (not shown). Additionally, after lag 0 (i.e., after central SSD date) a (weaker) tropopause wave source occurs, which shows clear upward and poleward wave propagation in both datasets, and in ERA-20C there is even an indication of
475 anomalous downward propagation (meaning either weaker upward wave propagation or weak downward wave propagation). Note that the potential downward propagation from the tropopause wave source could also be a consequence of wave reflection near the tropopause. The presence of the tropopause wave source following SSDs preceded by surface wave source is consistent with recent studies (e.g., Birner and Albers, 2017; de la Cámara et al., 2019).

  SSDs that are not preceded by neither tropopause nor surface wave source (Figs. 6d, 7d; though in ERA-20C these SSD
events can be preceded by $k = 2$ wave source as mentioned above) do not indicate any significant $k = 1$ wave propagation from troposphere to stratosphere in ERA-20C, but weak and discontinuous $k = 2$ wave propagation is present in the model

---

[3]SSDs/SSWs are considered explosive events, which are highly nonlinear; however, weaker anomalous behaviour in the troposphere and stratosphere (such as positive wind anomaly prior to an SSW/SSD) can be consistent with linear dynamics.

(consistent with weak wave sources there). In ERA-20C (as mentioned above) there is a weak but significant $k = 2$ wave source present at the tropopause (but there is no surface wave source) for SSDs not preceded by $k = 1$ wave source (not shown), which may be the cause of the $k = 2$ wave propagation found within the stratosphere, though a direct link to the $k = 2$ tropopause wave source is not clear. This means that these waves could also be generated via internal stratospheric dynamics, such as self-tuning resonance or downscale cascade[4]. The number of SSDs that are not preceded by any wave source is large ($\sim 50\%$) in the model, but smaller ($\sim 25\%$) in ERA-20C (see also Table 1). The exact reasons for this difference are currently unknown and require further studies. If we further exclude the SSDs preceded by the tropopause wave source events (located in the lower stratosphere), it leaves only $\sim 30\%$ of SSDs with a tropospheric precursor in both datasets (surface wave source events; see also Table 1). This suggests that extreme tropospheric wave forcing is not always necessary for producing SSD events, consistent with, e.g., Plumb (1981); Scott and Polvani (2004); Matthewman and Esler (2011); Sjoberg and Birner (2014); de la Cámara et al. (2019).

When averaging over all SSD events regardless of the wave source (Figs. 6a, 7a) we recover a combination of all of the above discussed cases with the model being dominated by SSDs not preceded by any wave source and SSDs preceded by surface wave source, whereas in ERA-20C a combination of SSDs preceded by surface and tropopause wave sources dominates. Here note that a combination of composites over the tropopause and surface wave source events provides a similar picture to the SSDs preceded by both the surface and tropopause wave source events, which have been omitted from Figs. 6, 7 for clarity. Note that an average over all SSDs somewhat recovers the differences between the model and ERA-20C found in section 4 (Figs. 4c, 5c), where a lagged relationship between the surface and tropopause wave sources was found in ERA-20C (similarly in Fig. 7a), but a more simultaneous relationship between the two was present in the model (similarly in Fig. 6a). It also shows a weak wave propagation back to the surface from the tropopause wave source at positive lags in ERA-20C (Fig. 7a), which is not seen in the model (Fig. 6a).

The above results also explain why meridional heat flux at 100 hPa, a proxy for vertical wave propagation (and vertical EP flux), might not be a good measure of the tropospheric wave forcing. This is because the upward wave propagation at 100 hPa could originate in the troposphere (at the surface), at the tropopause (lower-stratosphere), or even represent internal stratospheric dynamics. While 100 hPa heat flux increases the number of stratospheric events preceded by any precursor by about 30% (compared with the tropospheric, i.e., 700 hPa heat flux index only; e.g, White et al., 2019), its origin is unclear and could also lead to non-robust tropospheric response following the SSD/SSW event (see below). Here we have increased the number of SSDs preceded by any wave source event, by defining surface and tropopause wave source events, which has not been explored before. This means that if we consider surface wave source alone, only about a third of the SSDs are preceded by a wave source event, whereas including the tropopause wave source events increases the number of SSDs preceded by any wave source event by ∼30% in ERA-20C and by ∼13% in the model (see also Table 1). Another ∼10% in the number of SSDs preceded by precursory events can be gained by including SSDs preceded by both wave sources, though their impacts are less clear (not shown). While the percentage of SSDs preceded by the tropopause wave source in the model may seem

---

[4]The downscale cascade could be explored via similar methods as upscale cascade in the present study, though further examination of these mechanisms is left for future work.

small ($\sim$ 13%), the percentages in ERA-20C are much larger ($\sim$ 30%) and comparable to the percentages for the surface wave source ($\sim$ 30%), thus making the tropopause wave source a potentially important precursor for the SSDs in the real atmosphere (i.e., as important as the surface wave source). Recall that here the threshold for wave sources was lower than in the previous studies, thus more events may have been identified.

### 5.2.3 Downward impact after SSW events

The zonal mean zonal wind anomalies following the SSD events preceded by the tropopause and surface wave source events show strong deceleration in the stratosphere, which is robust across datasets (Figs. 6, 7), however surface impact is less clear. Fig. 6b,e shows an indication of a weak signal in the troposphere around lags 20 to 30 days following an SSD event, however Fig. 7b,e shows no tropospheric signal. Even though there is some surface signal present following SSD events, it is more common to find surface impact following SSW events, i.e., following a reversal of the stratospheric winds from westerlies to

easterlies. Therefore, compositing over the SSW events (Figs. 8, 9), where lag 0 is the first day when the stratospheric zonal flow reverses (around an SSD event), yields different (opposite) responses in the zonal flow in the troposphere for the SSWs preceded by the tropopause wave source (Figs. 8g,h,i, 9e,f) compared with SSWs preceded by the surface wave source (Figs. 8j,k,l, 9g,h).

For the SSWs preceded by the tropopause wave source the model (Fig. 8g; robust) and ERA-20C (Fig. 9e; less robust)

show wind deceleration in the 40-60°N latitudinal band at lags 15-25 days in the model and at lags 20-30 days in ERA-20C following an SSW event. Note that this downward impact can persist for over 20 days (lags 10-30 days and more; not shown), but is the most pronounced between the given lags. In the model this suggests an equatorward shift of the tropospheric jet stream (merged subtropical-extratropical jet stream), however in ERA-20C this may indicate different processes, since the subtropical and extratropical jet stream can be separated and lie in different locations compared with the model. Therefore, in

ERA-20C this may indicate: (i) a weaker eddy-driven (extratropical) jet stream; (ii) an equatorward jet shift of the subtropical jet stream, or (iii) a better split between the subtropical and extratropical jet streams. Note that at later lags (Figs. 8h,i, 9f) there is an indication of the tropospheric zonal flow deceleration further poleward. A further examination of the local responses (in ERA-20C) to the SSWs preceded by the tropopause wave source (Figs. S1, S2 third row, right column) reveals that there is a weakening of the Pacific jet stream, as well as a negative NAO signal in the Atlantic (equatorward extratropical jet shift there).

While the changes in the Pacific are consistent with the changes related to the tropopause wave source alone (Figs. S1, S2 fifth row, right column), the changes in the Atlantic are a response to SSWs. These results suggest that the zonal mean response is likely dominated by the responses in the Pacific (accompanied by the responses in the Atlantic). Further understanding of the local responses is left for future work.

Following the SSWs preceded by the surface wave source in the model (Fig. 8k) and in ERA-20C (Fig. 9g) there is a robust

downward impact at lags 30-40 days (model) and 20-30 days (ERA-20C) with tropospheric zonal flow deceleration polewards of 60°N, which is somewhat persistent (though less robust) also at other lags (e.g., Figs. 8l, 9h). In ERA-20C there is a robust acceleration of the zonal flow in the 40-60°N latitudinal band (Fig. 9g), i.e., the response to the SSWs preceded by the surface wave source is in the opposite sense to the response to the SSWs preceded by the tropopause wave source. As above, we can

gain further insight into the zonal mean picture in ERA-20C by examining the local response (Figs. S1, S2). This reveals that the surface wave source events are related to stronger jet stream in the Pacific regardless of the SSWs (compare Figs. S1, S2 fourth and sixth rows, right columns), however a response in the Atlantic is obscured by the SSWs, thus no clear response is found there. Again, this suggests that the zonal mean response is dominated by the responses in the Pacific. Note that the Pacific also shows a persistent signal at negative lags (i.e., prior to SSWs; Figs. S1, S2 right column, fourth row).

The results for the downward impact on zonal flow following SSWs thus show that this impact is different between the SSWs preceded by the tropopause wave source events and those preceded by the surface wave source events. The downward impact of SSWs preceded by the tropopause wave source events indicate the tropospheric zonal flow deceleration in the 40-60°N latitudinal band. Even though the downward impact occurs in a similar latitudinal band, the implications for the tropospheric jet stream changes may differ, e.g., equatorward shift of the tropospheric jet stream in the model versus weakening of the tropospheric jet stream in ERA-20C. On the other hand, the downward impact of SSWs preceded by the surface wave source events indicates tropospheric zonal flow deceleration further poleward (poleward of 60°N), and its acceleration in ERA-20C in the 40-60° latitudinal band. These results show that there are notable differences between the SSW response to different types of wave forcing in both datasets. This suggests that care must be taken when using indices (as also mentioned above), such as 100 hPa heat flux, since the downward impact (that we would ultimately like to predict) might be different depending on the origin of the waves at 100 hPa level (i.e., wave originating at the tropopause or at the surface).

The above results also suggest that in ERA-20C the zonal mean response is dominated by the response in the Pacific, which could be a consequence of the dynamics related to the wave sources themselves rather than the response to SSWs.

The tropospheric response to the SSWs that are not preceded by any wave source is less clear, i.e., somewhat different between the model and ERA-20C (Figs. 8d,e,f, 9c,d; see also Figs. S1, S2 second row). Similarly, there is also a robust but somewhat different downward impact seen in an average over all SSWs (regardless of the wave source; Figs. 8a,b,c, 9a,b; see also Figs. S1, S2 first row). Thus, the general (following all SSWs) downward impact in ERA-20C at later lags (35-45 days; Fig. 9b) shows zonal flow deceleration in the 60-80°N latitudinal band consistent with all studied cases (see Fig. 9d,f,h). On the other hand, the general downward impact in the model is first in the 50-70°N latitudinal band (lags 15-25 days; Fig. 8a), and at later lags (40-50 days; Fig. 8c) it is dominated by the response to SSWs not preceded by any wave source (Fig. 8f), i.e., an equatorward shift of the tropospheric zonal mean zonal wind. Locally, there is an indication of the negative NAO in ERA-20C following all SSW events (Figs. S1, S2, right column, first row), which is likely dominated by the signal from the SSWs preceded by the tropopause wave source events.

## 6 Conclusions

Recent work has identified a planetary wave source just above the tropopause, both in a climatological sense on the poleward side of the subtropical jet stream (Birner et al., 2013), as well as transiently on the poleward side of the extratropical jet stream, preceding the SSD events (de la Cámara et al., 2019). This study has examined the dynamical origins of the tropopause planetary wave source on the poleward side of the extratropical jet stream and its impacts on two-way stratosphere-troposphere

coupling. A better understanding of the tropopause wave source and its impacts may provide additional precursors to stratospheric events (SSDs, SSWs), potentially leading to a better prediction of the strong stratospheric events and their downward impact.

By analysing an idealised, mechanistic (dry dynamical core) general circulation model, and ERA-20C quasi-reanalysis, we have shown that the tropopause wave source can occur through upscale cascade (Figs. 2, 4c,d, 5c,d) as well as through wave decay (resulting in an apparent wave source; Figs. 1, 4e, 5e) in both datasets. While there are cases where only upscale cascade or only wave decay exist, they are more commonly occurring together, further amplifying the wave source signal at the tropopause. When the tropopause wave source occurs, the planetary waves then propagate out of the wave source region in all directions (upwards, equatorwards, polewards and even downwards; Figs. 6c, 7c), which can affect the atmospheric dynamics in the troposphere as well as stratosphere. As the waves propagate vertically into the stratosphere where they break and dissipate, they can decelerate the westerlies in the polar vortex (as indicated in Figs. 6c, 7c), which can lead to SSD and/or SSW events.

While only a small fraction of all tropopause wave source events ($\sim$4% in the model, $\sim$10% in ERA-20C; Table 1) lead to SSD/SSW events (Figs. 6b, 7b), they overall help increasing the number of SSDs/SSWs preceded by any wave source event (i.e., in addition to the surface wave source events), thus increasing the number of stratospheric events with a precursory wave source event, which is important for their predictability. Additionally, the SSW events preceded by the tropopause wave source have a similar downward impact in both datasets - there is a deceleration of the tropospheric zonal flow in the 40-60°N latitudinal band $\sim$15-30 days after the SSW event (Figs. 8g, 9e). In ERA-20C there is also a robust local response in the Atlantic, leading to equatorward shift of the jet stream (negative NAO) there. These results further suggest potential for predictability.

A comparison with the surface wave source events and their impact on the two-way stratosphere-troposphere coupling (Figs. 6e,f, 7e,f) reveals that a (weaker) tropopause wave source occurs following the surface wave source and SSD/SSW events (see also Birner and Albers, 2017; de la Cámara et al., 2019) as well as that the waves originating at the surface wave source largely sink in the upper troposphere, of which some make it into the stratosphere where they break (dissipate) and decelerate the zonal flow. When a tropopause wave source occurs following SSDs preceded by a surface wave source, there is also an indication of upward planetary wave propagation from this wave source, which could also be an indication of a positive feedback between a wave near the tropopause and a wave along the polar vortex. The downward impact following SSWs preceded by the surface wave source event (Figs. 8k, 9g) is in the opposite sense to the response following SSWs preceded by the tropopause wave source events, and thus the zonal flow deceleration occurs further poleward (poleward of 60°N) in both datasets, though at different lags (20-30 days in ERA-20C; 30-40 days in the model). Unlike for the SSWs preceded by the tropopause wave source events, no clear NAO signal in the Atlantic was found following SSWs preceded by the surface wave source.

The opposite signal in the tropospheric zonal flow following the different wave source events (i.e., at the surface versus at the tropopause) could be related to different tropospheric dynamics that is related to each of the wave source events. For example, during the surface wave source events there is a strong presence of the planetary waves within the troposphere prior to (and during) an SSW event, which can help accelerating the extratropical jet stream (especially in the Pacific in ERA-20C).

The presence of the planetary waves can also act to obscure the response of the synoptic waves to SSWs, which could be the cause for a weak response in the Atlantic. On the other hand, the absence of extreme planetary waves in the troposphere prior to SSWs preceded by the tropopause wave source events allows a wave-mean flow interaction including synoptic waves alone, thus potentially having opposite impact on the tropospheric extratropical jet stream (synoptic and planetary waves tend to have opposing impact on the mean flow; see, e.g., Hoskins et al., 1983). This can also lead to a stronger response of the synoptic waves to the SSWs (weaker interactions with the planetary waves), consequently leading to clearer equatorward shift of the jet stream (e.g., in the Atlantic in ERA-20C, in a zonal mean in the model).

The very different signal in tropospheric zonal flow following SSWs preceded by surface versus tropopause wave source events in ERA-20C and in the model as well as the presence of two different types of wave source events, also suggest that care must be taken when using indices, such as 100 hPa heat flux (see also de la Cámara et al., 2017). This is because: (i) the waves occurring at 100 hPa can be excited at the surface or at the tropopause (shown here), or even internally within the stratosphere (e.g., Plumb, 1981); and (ii) the SSWs preceded by the tropopause wave source lead to a downward impact (zonal flow deceleration) in 40-60°N latitudinal band, whereas SSWs preceded by surface wave source lead to an opposite surface signal with zonal flow deceleration further poleward.

Furthermore, we have also shown that the polar vortex preconditioning, i.e., strengthening of the polar vortex prior to SSD events (which can also be used as one of the precursors to SSDs; e.g., Jucker and Reichler, 2018) could be different between the SSDs preceded by tropopause and surface wave source events. This means that SSDs preceded by the tropopause wave source events are generally related to weaker zonal mean zonal wind strengthening (i.e., negative PV gradients) prior to SSDs (weaker polar vortex preconditioning is necessary), whereas SSDs preceded by surface wave source events are related to significant strengthening of the zonal flow prior to SSDs (i.e., stronger polar vortex preconditioning is necessary).

In summary, this study has addressed the dynamical origin of the tropopause wave source and analysed its impact on the two-way stratosphere-troposphere coupling. While this work has focused on the dynamical origins of the tropopause wave source, other potential wave sources with, e.g., a diabatic (via latent heat release) origin could be explored in the future, by, e.g., employing a hierarchy of models approach (e.g., Hoskins, 1983a; Held, 2005). Also, while local responses to SSWs preceded by different wave source events have briefly been mentioned in the present study, a better understanding of the local responses is also necessary, as it could provide further insight into the tropospheric dynamics around the SSW events. Further work is also necessary to test whether the tropopause wave source could be used as one of the predictors of the SSD/SSW events and their downward impact in, e.g., the subseasonal-to-seasonal model datasets (Vitart et al., 2017; Pegion et al., 2019).

*Code availability.* The model code is available on GitHub (https://github.com/lukelbd/gfdl-fms). The parameters are specified in section 3.1 and in Polvani and Kushner (2002); Gerber and Polvani (2009). Other scripts are available upon request.

*Data availability.* ERA-20C data was obtained from the European Centre for Medium Range Forecasts website (https://apps.ecmwf.int/datasets/data/era20c-daily/levtype=pl/type=an/).

*Author contributions.* LB performed the model experiment and its analysis. TB and LB analysed ERA-20C data. The figures were prepared by LB and TB, and the first draft of the manuscript was prepared by LB, which was further improved by TB for the final version.

*Competing interests.* The authors declare that they have no competing interests.

*Acknowledgements.* We thank two anonymous reviewers for their detailed and constructive comments that helped improve the original manuscript. We also thank Luke Davis for the help with the model and David Thompson for helpful discussions. This study was funded by the National Science Foundation Grant number AGS-1643167.

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

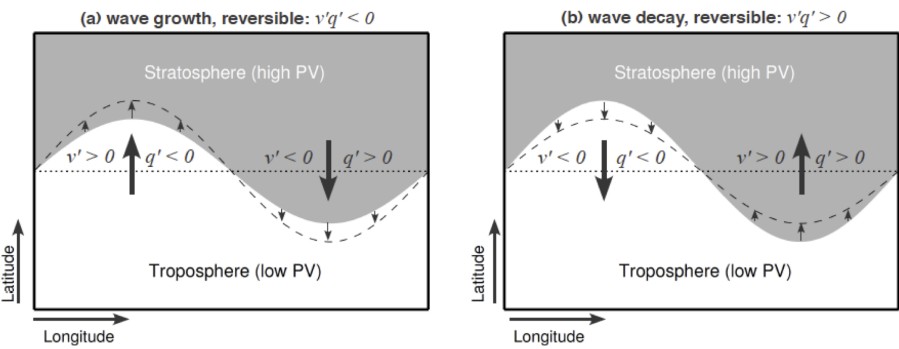

**Figure 1.** A schematic of a reversible wave growth (a) and decay (b) at the tropopause in a horizontal plane. Grey shading represents high PV air (from the stratosphere; PV also generally increases towards the pole), no shading (white) represents low PV air (from the troposphere; PV also generally decreases towards the Equator). Arrows denote the movements of the air as the wave grows or decays; $v'$ represents meridional movements; $q'$ represents changes in the PV following the meridional movements. Recall that $[v'q']$ is EP flux divergence (only for the QG dynamics). For a detailed description of the schematic see text.

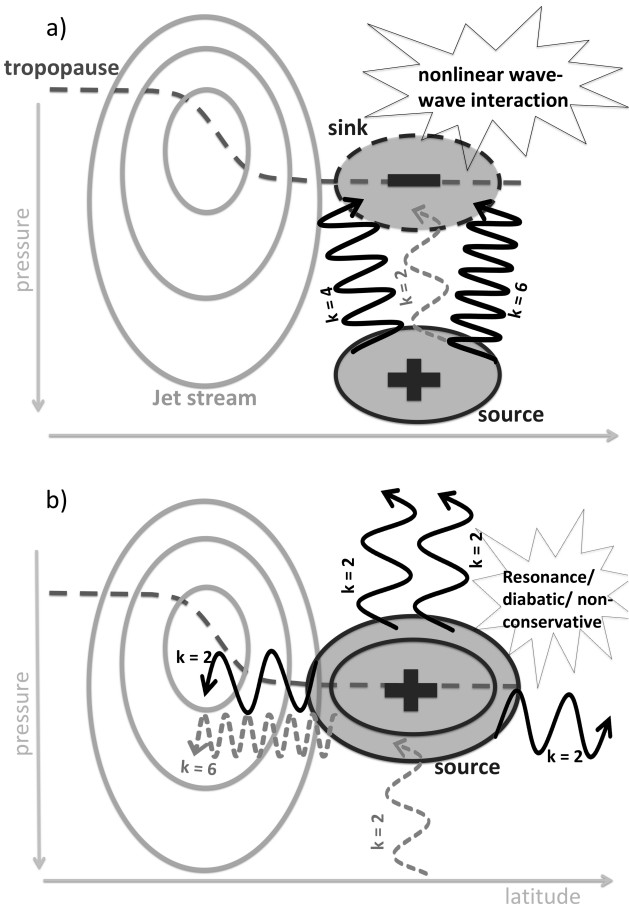

**Figure 2.** A schematic of upscale cascade at the tropopause in a vertical cross section: (a) smaller scale waves (e.g., $k = 4, 6$) show a sink of wave activity (i.e., EP flux convergence) at the tropopause, resulting in nonlinear wave-wave interactions; (b) an enhanced wave source appears in larger scale waves (e.g., $k = 2$), caused by upscale cascade, which can be accompanied by resonance, non-conservative or diabatic effects, further amplifying the wave source. For detailed description of the processes see text. Grey solid contours represent zonal mean zonal wind marking the jet stream; black solid (with plus sign in the middle) and dashed (with minus sign in the middle) ellipses represent wave sources ($\nabla \cdot \mathbf{F} > 0$) and sinks ($\nabla \cdot \mathbf{F} < 0$), respectively (the number of contours signifies the strength of $\nabla \cdot \mathbf{F}$); thick dark grey dashed line represents tropopause; solid black wiggly arrows (number of wiggles corresponding to the wave's wavenumber $k$, as labeled) represent the waves (and their propagation direction) that have the most importance at that stage in the process; grey dashed wiggly lines correspond to waves that may be present at that stage of the process but are not necessary. Note that $k = 2$ upward propagating wave represented by grey dashed line in both panels signifies a presence of a $k = 2$ wave that can interact with the waves generated via upscale cascade, leading to a potential resonant behaviour.

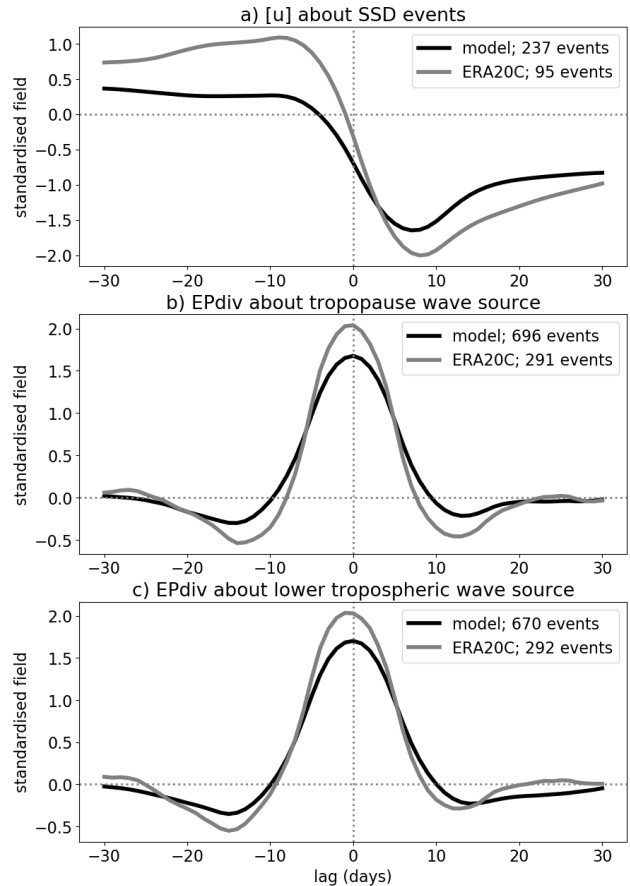

**Figure 3.** Index definitions shown through lag-composites of the relevant standardised quantities: (a) zonal mean zonal wind at 10 hPa (averaged between 45 and 75°N) about SSD events, (b) EP flux divergence ($k = 2$ for the model, $k = 1$ for ERA-20C) at tropopause about the tropopause wave source events, and (c) EP flux divergence ($k = 2$ for the model, $k = 1$ for ERA-20C) in the lower troposphere about the surface wave source events. Note that EP flux divergence was averaged between 40 and 60°N in the model and between 45 and 75°N in ERA-20C. Lag zero is the date when the index maximises (i.e. index central date). The lines denote different datasets: black lines for the model and grey lines for ERA-20C. Note that the data was smoothed by 10-day running mean before plotting.

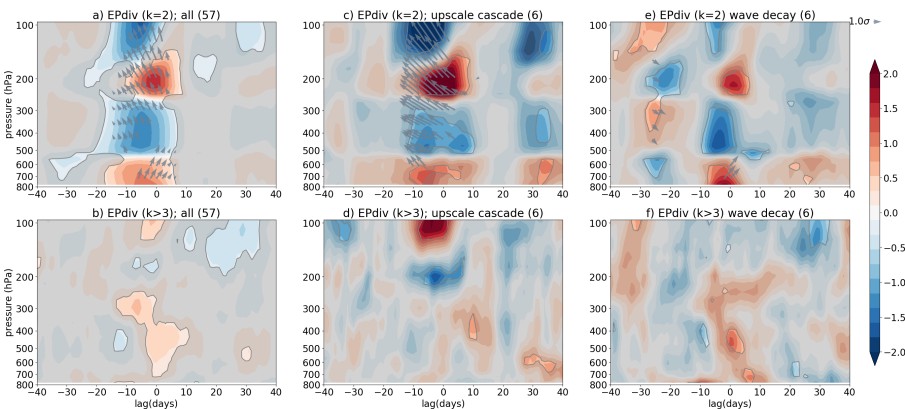

**Figure 4.** Composite analysis (lag-pressure for data averaged between 40 and 60°N) demonstrating the upscale cascade mechanism (based on a few subjectively identified events) (c,d) and wave decay (e,f) mechanisms, as well as a composite over all wave source events (a,b) preceding SSDs that were objectively identified. Top row (a,c,e) shows standardised $k = 2$ EP flux divergence anomalies (shading) and standardised EP flux anomalies (grey arrows), whereas bottom row (b,d,f) shows standardised EP flux divergence anomalies of the synoptic scale waves ($k \geq 4$). The arrows denote average *meridional* wave propagation direction (left-tilt: equatorward; right-tilt: poleward) and its magnification within the chosen latitudinal range at specified pressure/lag (i.e., not propagation in time), but do not imply actual size of the EP fluxes or the propagation out of the boundaries of the latitudinal range. Grey shading masks out data that are not significant at 95% level. EP fluxes (arrows) are only shown for values exceeding 95% significance level. The numbers in brackets denote the number of events in each composite. Data are from the model.

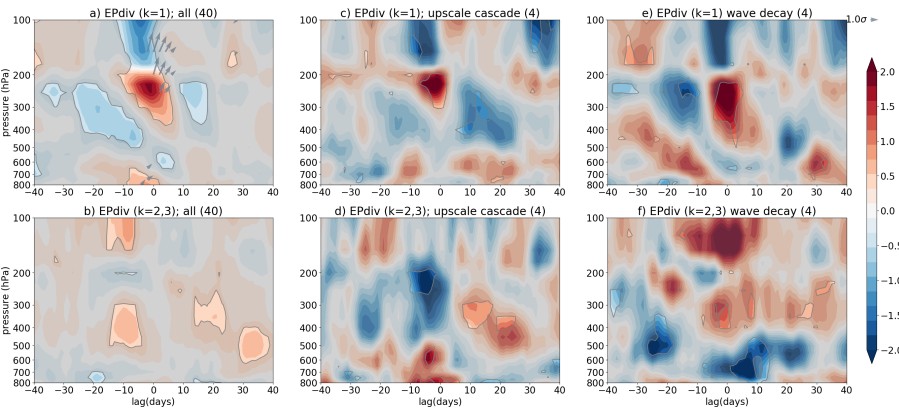

**Figure 5.** Similar to Fig. 4 but for ERA-20C data. Note that here top row represents the same quantities for $k = 1$ waves and bottom row represents the same quantities for $k = 2, 3$ waves instead of synoptic waves. Data are averaged between 45 and 75°N.

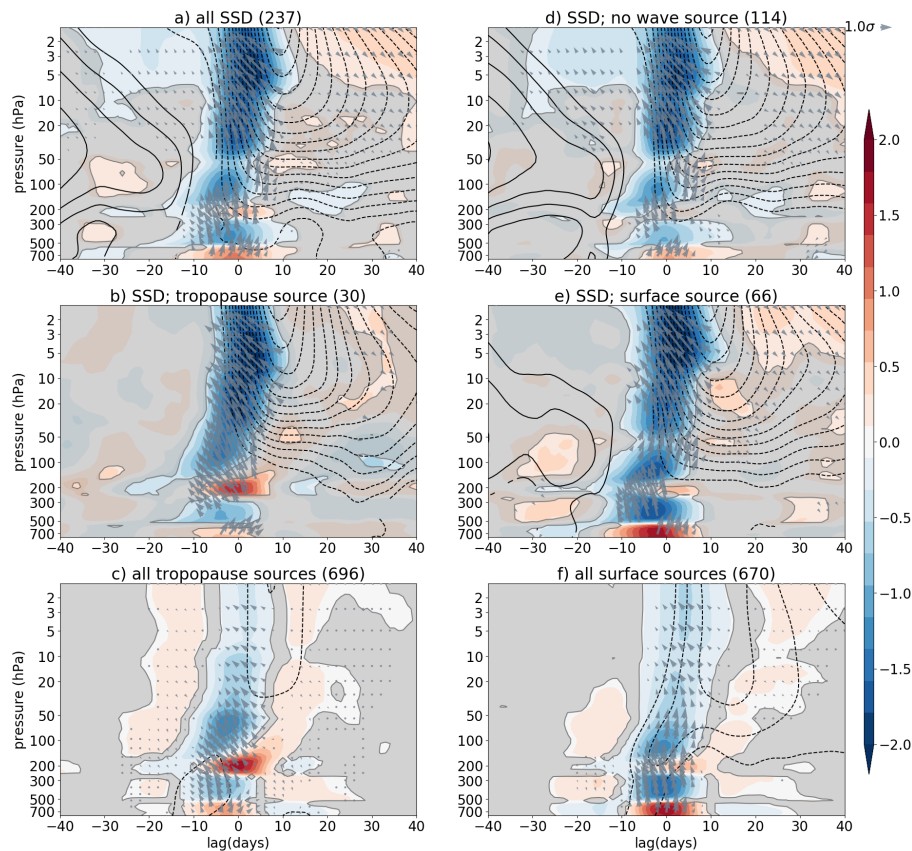

**Figure 6.** Composite analysis (lag-pressure for data averaged between 40 and 60°N) showing $k = 2$ standardised EP flux divergence anomalies (shading), $k = 2$ standardised EP flux anomalies (arrows), and standardised zonal mean zonal wind anomalies (contours; contour interval is 0.1, i.e. ...,-0.15, -0.05, 0.05, 0.15,...). The composites are shown for (a) all SSD events, (b) SSD events preceded by $k = 2$ tropopause wave source, (c) all $k = 2$ tropopause wave source events, (d) SSD events not preceded by $k = 2$ wave source events, (e) SSD events preceded by $k = 2$ lower-tropospheric wave source events, and (f) all $k = 2$ lower-tropospheric wave source events. (a,b,d,e) are centred around SSD events, whereas (c,f) are centred around wave source events. Grey shading masks out data that are not significant at 95% level. EP fluxes (arrows) and zonal mean zonal wind anomalies (contours) are only shown for values exceeding 95% significance level. As in Fig. 4, the arrows denote average *meridional* wave propagation direction (left-tilt: equatorward; right-tilt: poleward) and its magnification within the chosen latitudinal range at specified pressure/lag (i.e., not propagation in time). Numbers in brackets denote number of events in each composite. Data are from the model.

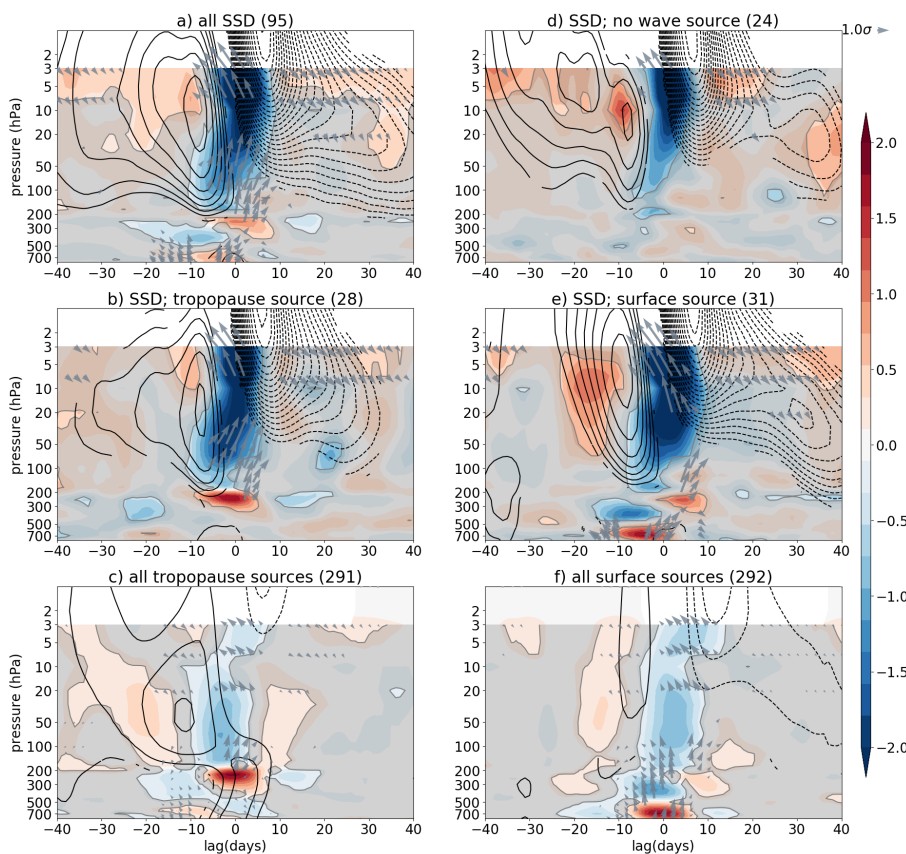

**Figure 7.** Similar to Fig. 6 but for ERA-20C data. Note that here the $k = 1$ standardised EP flux divergence and standardised EP fluxes (shading and arrows, respectively) are shown (and $k = 1$ wave source events were identified). Data are averaged between 45 and 75°N.

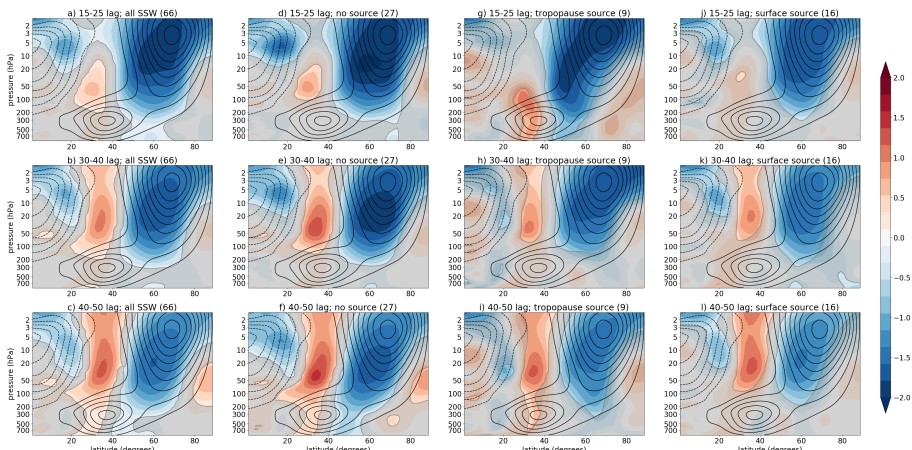

**Figure 8.** Composite analysis of downward impact in zonal mean zonal wind (latitude-pressure vertical cross section) averaged between the lags 15 and 25 days (top row), 30 and 40 days (middle row), and 40 to 50 (bottom row) following (a,b,c) all SSW events, (d,e,f) SSWs not preceded by any wave source events, (g,h,i) SSWs preceded by tropopause wave source, and (j,k,l) SSWs preceded by surface wave source. The figure shows standardised zonal mean zonal wind anomalies (shading) and zonal mean zonal wind climatology (contours; contour interval is 5 m s$^{-1}$ with 0$^{th}$ contour omitted for clarity, i.e. ...,-10, -5, 5, 10,...). Grey shading masks out data that are not significant at 95% level. Numbers in brackets denote number of events in each composite. Data are from the model.

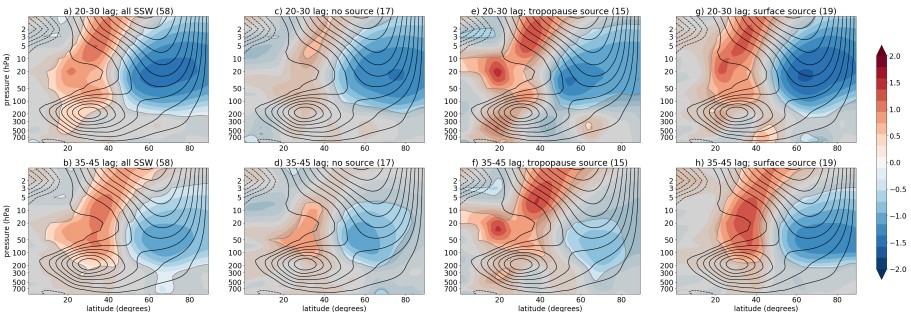

**Figure 9.** As in Fig. 8, but for ERA-20C data. Here composites are shown for zonal mean zonal wind (latitude-pressure vertical cross section) averaged between the lags 20 and 30 days (top row), 35 and 45 days (bottom row) following (a,b) all SSW events, (c,d) SSWs not preceded by any wave source events, (e,f) SSWs preceded by tropopause wave source, and (g,h) SSWs preceded by surface wave source.

**Table 1.** Total number of events (and percentage of all events) related to planetary wave sources and SSDs (as labeled). The numbers of events are also listed in Figs. 3, 6, 7.

| events / all events | model | ERA-20C |
|---|---|---|
| SSDs with tropopause wave source / all tropopause wave source | 30 / 696 (4.3%) | 28 / 291 (9.6%) |
| SSDs with surface wave source / all surface wave source | 66 / 670 (9.9%) | 31 / 292 (10.6%) |
| SSDs with tropopause wave source / all SSDs | 30 / 237 (12.7%) | 28 / 95 (29.5%) |
| SSDs with surface wave source / all SSDs | 66 / 237 (27.8%) | 31 / 95 (32.6%) |
| SSDs without wave sources / all SSDs | 114 / 237 (48.1%) | 24 / 95 (25.3%) |
| SSDs with both wave sources / all SSDs | 27 / 237 (11.4%) | 10 / 95 (10.5%) |