# Peer review of "Tropopause-level planetary wave source and its role in two-way troposphere-stratosphere coupling"

_Weather and Climate Dynamics, 2020_

## Referee Comment (RC1) · Anonymous Referee #1 · 21 Jun 2020

Review of " Tropopause-level planetary wave source and its role in two-way" by Boljka and Birner

The authors analyze the possible forcing mechanisms for a planetary wave source near the tropopause that subsequently propagates upwards. The authors identify two different mechanisms: nonlinear wave-wave interactions and subsequent resonance, and also transient wave decay. They find a more robust downward impact for SSW preceded by a tropopause wave source event. This paper has many interesting results, and while there are some points the authors need to clarify, it is very likely that the paper will be suitable for publication after revisions which are best classified as somewhere

between major and minor.

Major comments: 1. The authors argue that the upscale cascade is then followed by resonance, but little evidence is provided for resonance actually occurring, nor is resonance in the present context defined (the relevant section in Vallis doesn't help). There is a brief statement that e.g. the magnitude of EP flux divergence of wave 2 exceeds the convergence for synoptic wavenumbers (line 284-285), however this does not necessarily mean resonance is occurring. While a sudden change of EPFD would imply a change in wave activity if non-conservative processes are not present (equation 7.23a of Vallis 2006), this relationship is derived under specific limitations. However more generally (finite amplitude and non-QG), EPFD is not related directly to wave activity and thus not there is no expectation that it should be conserved. That is, a negative EPF in a given wavenumber range does not need to be balanced by positive EPFD in a different wavenumber range in a turbulent cascade with dry dynamics only. Furthermore, even if EPF was directly related to conserved wave activity, diabatic and other such non-conservative processes can create wave activity, and hence it is impossible given the diagnostics shown to determine whether the increase in planetary waves is indeed due exclusively to triad interactions as the authors suggest, or other processes. Overall there is no need for EPFD to be conserved in general, let alone locally.

Second the terminology is a bit confusing, as the authors refer to resonance here not in the context used most often in stratospheric dynamics (the way e.g. Plumb 2010 described it, or the way Matthewman and Esler 2011 have in mind). This distinction could be made somewhat clearer.

Finally, there is an entire literature in turbulence community of physics on a concept called wave turbulence. I am certainly not an expert on this topic, but I have had interactions with people from the turbulence community, and there is an entire book on the wave turbulence regime where it is meaningful to focus on specific triad interactions but to ignore the entire spectrum of possible interactions (Zakharov et al 2012). In short, the regime of any turbulent system can be characterized by the Reynolds number

Re. Re is defined as the ratio of the dissipation time $L^2/\nu$ due to viscosity and the inertial time $L/V$. The inertial time characterizes the generation by triad interactions of other Fourier harmonics out of velocity fluctuations with characteristic scale $L$ and characteristic velocity $V$, often referred as "eddies", that are injected by the force. If Re«1 then viscosity dissipates the eddy before non-linear interactions can produce other eddies and the flow is effectively linear. In contrast, if Re»1 then the initial eddy injected by the forcing generates eddies of comparable, yet different, scale before any dissipation occurs. These eddies in turn generate by non-linear interaction other eddies with scale which is already comparable with theirs.

For the present study, Re can be defined as $Re=V\tau/L$ where $\tau$ is the characteristic time of decay of fluctuations due to frictional processes, $V$ is some metric of velocity, and $L$ is a metric of time. If Re«1 then the friction dissipates fluctuations created by the forcing before they can transfer appreciable energy to other modes by triad interactions. In contrast, if Re»1 the energy cascade would occur. Wave turbulence of the kind the authors have in mind (where a specific triad interaction can be studied ignoring all other possibilities) is only relevant when Re is less than "around" 1 (with the definition "around" very specific to the problem at hand). Regardless of the relevant value of Re an upscale cascade occurs, but a focus on the EPFD budget for individual triad interactions seems misplaced since there is no reason for EPFD to be conserved.

In short, unless the authors are able to bolster their results, I recommend deleting the claim that resonance is occurring. (I think this paper would still be a useful contribution without this claim.) At most, the possibility of resonance could be broached in the discussion section.

2. This is more a pet peeve than a major comment, but the authors cite several papers in the introduction claiming that "anomalous wave forcing is not always necessary for producing SSD events". However these previous papers define wave forcing events by 2 std deviations. A 1.95 std deviation wave forcing from the long-term mean is clearly "anomalous" to me, but would be classified by some of these papers as not particularly

noteworthy.

In this paper the authors use a lower threshold, and claim that there are still many SSD events without an anomalous wave forcing (line 443). However Figure 6d clearly shows an anomalous wave forcing both at the surface and also at the tropopause. Presumably the 0.75 std dev threshold used here just barely misses some events that end up being included in Figure 6d. While these events certainly don't have "extreme" wave flux, wave fluxes are anomalous, and the use of "anomalous" on line 443 is incorrect.

Here is a list of lines where "anomalous" should be replaced with "extreme": line 42, 443, 541 "extreme" could also be added to line 36 .

3. I found figure 8 and its accompanying discussion a bit under-developed. First of all, it is strange that the authors find no significant tropospheric impact when all SSW are composited (line 504-505). This seems contrary to dozens of published studies finding a downward surface impact from SSWs. Second, the tropospheric impact is much clearer when focusing on the Atlantic sector (i.e. the NAO) and not the zonal mean (dozens of studies on this too), and the analysis here would be much more convincing if in addition to (or instead of) the zonal mean figures the authors only composited winds in the Atlantic sector. In other words, instead of showing zonal mean zonal winds, please only show zonal winds averaged between, say, 300E and 20E. A map view figure might also help in interpreting the results, especially in discriminating among the three options listed on lines 478 to 481.

minor comments: line 99:The Held et al 2002 review paper and the recent study Garfinkel et al 2020 should be added here. More generally, it isn't clear to me that these forcings should lead to a lower tropospheric wave source per se, and not a wave source higher in the troposphere. For example, upper tropospheric diabatic heating due to baroclinic instability or land-sea contrast should directly affect upper tropospheric stationary waves. That being said, I agree that these factors are likely not directly forcing waves at the tropopause.

Line 113-114 I found this sentence difficult to parse. Please rephrase.

line 231: It is impossible to tell from this figure that the events are long-lived due to the 10 day smoothing filter applied. If it is important to emphasize the long-lived nature of the EP flux divergence, then please perhaps add a thin line for the (raw) non-filtered data for the model and quasi-reanalysis.

Figure 3a: please indicate the pressure level for [U] either in the caption or on the figure itself

Is time smoothing applied for figure 4 and 5 and subsequent figures, or is figure 3 the only figure with time-smoothing? Please clarify either way. The reason I ask is that the text near line 258/259 seems to imply a time separation in the "synoptic" vs. planetary EPFD, however no such time separation appears in figure 4 and 5 (rather the planetary and "synoptic" waves change essentially simultaneously)

Colorbar tick labels on figure 4,5, 6, etc.– please label the ticks symmetrically about zero.

Figure 4 and 6: I found the equatorward/poleward tilt of EPF arrows to be somewhat confusing. I first thought this reflected some sort of propagation backwards in time, which of course makes no sense, but then I reread and understood. I don't think the question of equatorward vs. poleward propagation is particularly important to this study.

Line 386- there is a positive stratospheric wind anomaly before the SSD in figure 6b, it is just weaker than in 6a,6e or 6e.

Lime 388 I don't understand this comment about a lack of preconditioning in figure 3a. Winds are clearly stronger than average before the SSD on figure 3a

Line 392- is this difference in [U] before the SSD between the "tropopause source" composite and the "surface source" composite actually statistically significant? I.e. a difference plot between panels b and e.

Technical comments Line 121 in *the* lower

Plumb, R. Alan. "Planetary waves and the extratropical winter stratosphere." The Stratosphere, Dynamics, Transport and Chemistry, Geophys. Monogr 190 (2010): 23-41.

Matthewman, N.J. and Esler, J.G., 2011. Stratospheric sudden warmings as self-tuning resonances. Part I: Vortex splitting events. Journal of the atmospheric sciences, 68(11), pp.2481-2504.

Held, I.M., Ting, M. and Wang, H., 2002. Northern winter stationary waves: Theory and modeling. Journal of climate, 15(16), pp.2125-2144.

Garfinkel, C.I., White, I., Gerber, E.P., Jucker, M. and Erez, M., 2020. The building blocks of Northern Hemisphere wintertime stationary waves. Journal of Climate, 33(13), pp.5611-5633.

Zakharov, V.E., L'vov, V.S. and Falkovich, G., 2012. Kolmogorov spectra of turbulence I: Wave turbulence. Springer Science & Business Media.

---

## Referee Comment (RC2) · Anonymous Referee #2 · 24 Jun 2020

This paper addresses an important subject in stratosphere-troposphere coupling, which is how weak vortex events are ultimately forced. In particular, the authors concentrate on the appearance of a wave source around the tropopause in contrast to most work considering surface or mid-tropospheric wave sources. It is good to see new ideas explored, and I am looking forward to seeing this published. However, there are a few issues as described below which need to be addressed.

Major comments:

1) My first major comment is about the model selection. a) The Held-Suarez/Polvani-Kushner model has been known to exhibit somewhat unrealistic dynamical behavior.

[Figure]

Adding topography has mostly fixed the jet position issue, but what is probably more important in this particular study is the generally too low tropopause. I do not know what starting the stratospheric setup at 200hPa instead of 100hPa as described does to this bias, and there is no mention of any model validation in the manuscript. For instance, it is not true that dry dynamical cores do not produce SSWs with k=1 topography. The authors cite Sheshadri et al (2015), which is the model setup they use, but there are various other, improved versions of the dynamical core which do produce SSWs with wave-one (or realistic) topography. b) Throughout the manuscript, there are important differences between the model and ERA-20C, but those are not critically discussed. For instance, Fig. 4c) shows synchronous wave sources at both the surface and the tropopause, whereas 5c) indicates a possible propagation from the surface (at -10d) to the tropopause (around 0) and then back to the surface (at 5-10d and later). This is similar for Figs. 6a) and 7a). Fig. 6b) shows a surface source in the tropopause composite, whereas 7b) does not. Also, while shortly discussed in the manuscript, the preconditioning of the stratosphere in Figs 6 and 7 has a very different structure in the model vs ERA-20C, and so does the zonal mean zonal wind anomaly in Fig. 8. In Table 1, some percentages are close to ERA-20C, some are not, but there is no discussion of the confidence in the model results.

I think these differences are qualitative and require some discussion. Or the use of a different model with more realistic behavior.

2) Another important comment is a missing clear acknowledgment that most SSDs do not have any of the two wave sources. I understand the subjective selection to be able to study "clean" examples, and they do reveal interesting physics. But I feel that the manuscript is missing an estimate of how important the studied mechanisms really are. Another point is that only 10% of the tropopause source events are also SSDs (4% in the model), so what is happening with the other 90%? Same is true for surface source events.

Minor comments:

L 28-29: "Here note ..." should probably be moved to line 24, where Charney and Drazin (1961) is already discussed.

L 31-32: This list of references is too long. It does not concentrate on the most important works but is not exhaustive either, and seems to mainly serve self-citation.

L 42: "suggest that an anomalous ..."

L 43: Same as L 31-32: too long but not exhaustive list of references. Also, shouldn't "Camara et al." read "de la Camara et al."?

L 102: "Note that the wave decay..." is not necessary as this is discussed immediately afterward.

L 113-14: "Therefore, even...": you just concluded that div(F) = 0, so where is the "significant increase"? I am sure you are trying to say something different, but this is confusing as written.

L 124: "(sink)": this is a bit confusing, as at first I thought this was meant to mean that the waves are sinking downward. Maybe move this into the parenthesis "(negative EP flux divergence, i.e. a sink)" or similar.

L 131: Note that the newly created k=2 wave can also cancel an existing k=2 wave if the phase is opposite. Try and be more careful when describing the triad interactions.

Paragraph 2.2: Note that while valid, the triad interactions can (and probably will) only convert waves partially, i.e. there is going to be partial dissipation and resonance, plus other effects, so a comment on the testability of this process would be welcome.

Paragraph 3.1: Maybe change title to "Model & Data"

L 164: 0 hPa is at infinity, so surely the model top is somewhere else?

L 193: 20-days -> 20 days

Paragraph 3.3: It is ok to subjectively select a few "clean" events, but there should be

an estimate of the relative importance of what you are filtering out, i.e. you select a few events which show your mechanism, but in the grand scheme of things, how important is that mechanism?

L 332: remove "(section 5)" as that's the very next thing.

L 363-365: This relates to some of the comments above: Do you have any interpretation as to why only 10/4% result in SSDs and what happens with the remaining 90/96%?

L 424: "into deep stratosphere": maybe change to "deep into the stratosphere"

L 439-440: Do the authors have any suggestion about how one could check for self-tuning or downscale cascade? Not necessary, but would be very helpful for future work.

L 444: Again, there are others with this idea

L 491-492: The difference between model and ERA-20C seems much larger than the difference between tropopause and source wave source events. Can you then really assert that there is a difference between tropopause and surface wave source in terms of downward impact?

L 515-516: "While cases..." there is something missing in this sentence.

L 517: occurs, the

---

## Author Comment (AC1) · 5 Aug 2020

**Response to Reviewer 1**

We would like to thank the reviewer for carefully reading the manuscript, and for their detailed and constructive comments that will ultimately help improving the original manuscript. Below are our responses to the reviewer, which will be implemented in the revised manuscript in the next stage. Note that we have not provided exact manuscript corrections at this point, but we have provided the intended changes in detail; all figures that were produced in response to the reviewer's comments are at the end of this document; the line numbers/figure references in the reviewer's comments refer to

the original manuscript. The reviewer's comments are in italics; our responses are in normal text.

*The authors analyze the possible forcing mechanisms for a planetary wave source near the tropopause that subsequently propagates upwards. The authors identify two different mechanisms: nonlinear wave-wave interactions and subsequent resonance, and also transient wave decay. They find a more robust downward impact for SSW preceded by a tropopause wave source event. This paper has many interesting results, and while there are some points the authors need to clarify, it is very likely that the paper will be suitable for publication after revisions which are best classified as somewhere between major and minor.*

**Major comments**

*1. The authors argue that the upscale cascade is then followed by resonance, but little evidence is provided for resonance actually occurring, nor is resonance in the present context defined (the relevant section in Vallis doesn't help). There is a brief statement that e.g. the magnitude of EP flux divergence of wave 2 exceeds the convergence for synoptic wavenumbers (line 284-285), however this does not necessarily mean resonance is occurring. While a sudden change of EPFD would imply a change in wave activity if non-conservative processes are not present (equation 7.23a of Vallis 2006), this relationship is derived under specific limitations. However more generally (finite amplitude and non-QG), EPFD is not related directly to wave activity and thus not there is no expectation that it should be conserved. That is, a negative EPF in a given wavenumber range does not need to be balanced by positive EPFD in a different wavenumber range in a turbulent cascade with dry dynamics only. Furthermore, even if EPF was directly related to conserved wave activity, diabatic and other such non-conservative processes can create wave activity, and hence it is impossible given the*

*diagnostics shown to determine whether the increase in planetary waves is indeed due exclusively to triad interactions as the authors suggest, or other processes. Overall there is no need for EPFD to be conserved in general, let alone locally.*

*Second the terminology is a bit confusing, as the authors refer to resonance here not in the context used most often in stratospheric dynamics (the way e.g. Plumb 2010 described it, or the way Matthewman and Esler 2011 have in mind). This distinction could be made somewhat clearer.*

*Finally, there is an entire literature in turbulence community of physics on a concept called wave turbulence. I am certainly not an expert on this topic, but I have had interactions with people from the turbulence community, and there is an entire book on the wave turbulence regime where it is meaningful to focus on specific triad interactions but to ignore the entire spectrum of possible interactions (Zakharov et al 2012). In short, the regime of any turbulent system can be characterized by the Reynolds number $Re$. $Re$ is defined as the ratio of the dissipation time $L^2/\nu$ due to viscosity and the inertial time $L/V$. The inertial time characterizes the generation by triad interactions of other Fourier harmonics out of velocity fluctuations with characteristic scale $L$ and characteristic velocity $V$, often referred as "eddies", that are injected by the force. If $Re \ll 1$ then viscosity dissipates the eddy before non-linear interactions can produce other eddies and the flow is effectively linear. In contrast, if $Re \gg 1$ then the initial eddy injected by the forcing generates eddies of comparable, yet different, scale before any dissipation occurs. These eddies in turn generate by non-linear interaction other eddies with scale which is already comparable with theirs.*

*For the present study, $Re$ can be defined as $Re = V\tau/L$ where $\tau$ is the characteristic time of decay of fluctuations due to frictional processes, $V$ is some metric of velocity, and $L$ is a metric of time. If $Re \ll 1$ then the friction dissipates fluctuations created by the forcing before they can transfer appreciable energy to other modes by triad interactions. In contrast, if $Re \gg 1$ the energy cascade would occur. Wave turbulence of the kind the authors have in mind (where a specific triad interaction can be studied*

*ignoring all other possibilities) is only relevant when $Re$ is less than "around" 1 (with the definition "around" very specific to the problem at hand). Regardless of the relevant value of Re an upscale cascade occurs, but a focus on the EPFD budget for individual triad interactions seems misplaced since there is no reason for EPFD to be conserved.*

*In short, unless the authors are able to bolster their results, I recommend deleting the claim that resonance is occurring. (I think this paper would still be a useful contribution without this claim.) At most, the possibility of resonance could be broached in the discussion section.*

We agree with the reviewer about the limitations of the QG EPFD framework, which is based on quasi-linear theory. However, despite these limitations the QG EPFD framework has been successfully used in the past for studying stratospheric and tropospheric dynamics. For these reasons we use it here, although more so in a qualitative than quantitative sense. We will include remarks on the limitations of the quasi-linear framework at the beginning of section 2. Note that in the model diabatic effects at the tropopause are not present (there is dissipation, but no diabatic source), but in the real atmosphere (though still unlikely) they can be present. This has been mentioned briefly in the methods section, but we will clarify this further in the text. As the reviewer suggests, we will discuss other options for amplified EPFD during upscale cascade, i.e. resonance, non-conservative and diabatic effects, which have also been added to the upscale cascade schematic - Fig. 1b below (revised Fig. 2 from the original manuscript). Therefore, as per reviewer's suggestion we will discuss resonance more carefully throughout the manuscript.

We thank the reviewer for providing detailed comments on turbulence theory and triad interactions. Note that in the present context we think of the upscale cascade mechanism as primarily representing wave-wave interactions of a finite number of large scale waves, which can be highly nonlinear but do not necessarily represent turbulent interactions across a quasi-continuous range of wave numbers. We will clarify this distinction in the text. We will also clarify the difference between the self-tuning resonance that occurs in the stratosphere and the one described here, when introducing potential resonance that follows upscale cascade (in section 2).

*2. This is more a pet peeve than a major comment, but the authors cite several papers in the introduction claiming that "anomalous wave forcing is not always necessary for producing SSD events". However these previous papers define wave forcing events by 2 std deviations. A 1.95 std deviation wave forcing from the long-term mean is clearly "anomalous" to me, but would be classified by some of these papers as not particularly noteworthy.*

*In this paper the authors use a lower threshold, and claim that there are still many SSD events without an anomalous wave forcing (line 443). However Figure 6d clearly shows an anomalous wave forcing both at the surface and also at the tropopause. Presumably the 0.75 std dev threshold used here just barely misses some events that end up being included in Figure 6d. While these events certainly don't have "extreme" wave flux, wave fluxes are anomalous, and the use of "anomalous" on line 443 is incorrect.*

*Here is a list of lines where "anomalous" should be replaced with "extreme": line 42, 443, 541 "extreme" could also be added to line 36.*

We will replace "anomalous" with "extreme" where relevant as suggested by the reviewer.

*3. I found figure 8 and its accompanying discussion a bit under-developed. First of all, it is strange that the authors find no significant tropospheric impact when all SSW are composited (line 504-505). This seems contrary to dozens of published studies finding a downward surface impact from SSWs. Second, the tropospheric impact is much clearer when focusing on the Atlantic sector (i.e. the NAO) and not the zonal mean (dozens of studies on this too), and the analysis here would be much more convincing if in addition to (or instead of) the zonal mean figures the authors only composited winds in the Atlantic sector. In other words, instead of showing zonal mean zonal*

*winds, please only show zonal winds averaged between, say, 300E and 20E. A map view figure might also help in interpreting the results, especially in discriminating among the three options listed on lines 478 to 481.*

We thank the reviewer for mentioning this issue - this allowed us to recognise an error in the code - it only applies to Fig. 8, which will be replaced by Figs. 2,3 below. The discussion around the Fig. 8 (especially in section 5.2.3; but also in conclusions and in the abstract) will be revised accordingly. We have also looked into local responses by plotting maps, which further revealed interesting points (see Figs. 4,5 below). However, since we now see a zonal mean downward impact and as we have map plots, which show weak impact in the Atlantic for surface wave source, we will not use the N. Atlantic zonal mean. The zonal mean picture has been well-established in the literature and as such provides easier comparison to previous work. Maps further explain some of its results. Here are the main points that we will now discuss in section 5.2.3, as shown in Figs. 2-5 below.

(i) Zonal mean figures (Figs. 2,3 below) reveal that the model (Fig. 2) and ERA-20C (Fig. 3) largely agree on the sign of the anomalies following SSWs preceded by the tropopause (Fig. 3e,f, 2g,h,i) and surface (Fig. 3g,h, 2j,k,l) wave source events. Therefore, despite the more robust downward impact for SSWs preceded by surface wave source events in ERA-20C, and a more robust response for SSWs preceded by tropopause wave source events in the model, there is a general agreement between the two datasets on the tropospheric zonal mean zonal wind response to SSWs preceded by different wave source events. The response to SSWs in both datasets reveals opposite signed anomalies of the zonal mean zonal wind following SSWs preceded by the tropopause and surface wave source events, with the latter having a negative anomaly further poleward (60-90N) than the former (40-60N). This is consistent with the original manuscript, where we already suggested that this could be a result of different impacts of planetary and synoptic waves on driving the tropospheric mean flow (and in the tropopause wave source case no extreme planetary wave source is found in the

troposphere prior to the SSW event, thus a different response makes sense). Also, the robust 'response' of the tropospheric winds to the surface wave source could be partly a result of the wave-mean flow dynamics in the troposphere, regardless of the SSW events (see text about map results below). This would mean that when there is a surface wave source present, the tropospheric dynamics behave in a similar way, whether there is an SSW or not (but only in the Pacific; SSWs seem to obscure the behaviour in the Atlantic - see below). We can now also recover downward impact in zonal mean zonal winds following all SSWs in both datasets as well (e.g. Fig. 3b, 2a,c).

(ii) Figs. 4,5 show map-composites over all SSW events (top row), SSWs without wave source event (second row), SSWs preceded by tropopause wave source (third row), SSWs preceded by surface wave source (fourth row), all tropopause wave source events regardless of SSWs (fifth row), and all surface wave source events regardless of SSWs (bottom row). The composites are shown for zonal wind at 950 hPa (U950) and geopotential height at 500 hPa (Z500), respectively, for 30 days prior to SSW or wave source event and for 30 days following the event in ERA20C. While the case for all SSWs picks up the strongest signal from all the cases studied here, it now clearly shows a strengthening of the Pacific jet stream at negative lags, and equatorward shift of the Atlantic jet at positive lags (Fig. 4), consistent with reviewer's remarks. Consistent with this, the Z500 (Fig. 5) shows stronger Aleutian low as well as Scandianvian blocking-like signal over Europe at negative lags, and negative North Atlantic Oscillation (NAO-)-like signal at positive lags. However, we find that it is the SSWs preceded by the tropopause wave source that give a robust response in the Atlantic following SSWs (third rows in Figs. 4,5), i.e. NAO- signal at positive lags in both U950 and Z500, where the latter also revealed Greenland-blocking-like signal (not shown). Also, at negative lags there is clear Scandinavian-blocking-like signal, which has been shown in previous studies to have preceded SSWs and NAO- events (e.g. Kautz et al. 2020). This case also shows positive U950 anomalies in the subtropical jet region, suggesting shifts in Hadley cell following SSWs as well. On the other hand, the SSWs preceded by surface wave source show no clear signal in the Atlantic, but instead show a very

robust signal in the Pacific, consistent with strengthening of the jet stream there both prior and after the SSW event, which is likely a consequence of the wave-mean flow interaction within the troposphere there (fourth and bottom row in Fig. 4 have similar signals in the Pacific). This is likely the case for the tropopause wave source in the Pacific as well (third and fifth rows in Fig. 4). In the Atlantic the surface wave source suggests an NAO- signal (bottom right panel in Figs. 4, 5), which is likely disturbed by the SSWs (fourth row right panel in Figs. 4,5). Here note that at positive lags the SSWs preceded by tropopause and surface wave sources have opposite signs in the Pacific (similar for all wave source events), which could be responsible for the opposing downward impact in a zonal mean as well. The results for SSWs that are not preceded by any wave source events are not clear. Note that while the results for the local surface response are interesting, we will keep their discussion to a minimum (e.g. in a supplement) as studying local impacts is beyond the scope of this study.

Also note that the downward impact in ERA-20C (average over all SSW events) may be different (slightly less robust) to downward impact in satellite-era reanalyses for two reasons: (i) ERA-20C is constrained by surface observations only; and (ii) ERA-20C has 50+ years of data more than satellite-era reanalyses (more events that can obscure the signal found in satellite-era). For further discussions on different reanalyses see, e.g., Gerber and Martineau (2018), Hitchcock (2019). We will mention these points in the text.

**Minor comments**

*line 99: The Held et al 2002 review paper and the recent study Garfinkel et al 2020 should be added here. More generally, it isn't clear to me that these forcings should lead to a lower tropospheric wave source per se, and not a wave source higher in the troposphere. For example, upper tropospheric diabatic heating due to baroclinic insta-*

*bility or land-sea contrast should directly affect upper tropospheric stationary waves. That being said, I agree that these factors are likely not directly forcing waves at the tropopause.*

We will add the references as suggested. Generally, baroclinic instability and land-sea contrasts play a role closer to the surface; we will clarify that these mechanisms are not excluded further up in the troposphere, though their direct impacts are unlikely at the tropopause.

*Line 113-114 I found this sentence difficult to parse. Please rephrase.*

We will rephrase this as: "Therefore, even if the EP flux divergence exceeds a set threshold and appears as though there is a wave source, this is merely representing a decay of a wave, and thus we will refer to it as an apparent wave source."

*line 231: It is impossible to tell from this figure that the events are long-lived due to the 10 day smoothing filter applied. If it is important to emphasize the long-lived nature of the EP flux divergence, then please perhaps add a thin line for the (raw) non-filtered data for the model and quasi-reanalysis.*

Fig. 6 below shows the composite over 10-day smoothed data (thick lines) and un-smoothed data (thin lines) as the reviewer suggested. The figure shows that un-smoothed EP flux divergence shows less persistence but there is still a peak of $\sim 2\sigma$ and $\sim 8+$ days persistence (i.e., EP flux divergence exceeding $0.75\sigma$ for over 8 days), as well as anomalously positive EP flux divergence persistence for an even longer period. This means that while the original Fig. (i.e. from smoothed data) does not necessarily show the persistence, persistent events still exist. We will rephrase the sentence "This shows that in both datasets there are long-lived (exceeding the $0.75\text{-}\sigma$ threshold for over 10 days) wave source events at the tropopause and in the lower troposphere (at surface), and that they have a similar evolution and standardised strength (peaking at $2\sigma$) when they occur." as (or along the lines of) "The figure shows wave source

events at the tropopause and at the surface, which are similar in strength (peaking at $2\sigma$) and persistence. These events appear long-lived (exceeding the 0.75-$\sigma$ threshold for over 10 days) due to a 10-day smoothing applied before compositing over all events. Thus, care must be taken in interpreting this persistence, even though there are individual long lived events in the dataset. Note that it is the 10-day mean wave forcing exceeding the threshold (e.g., $0.75\sigma$) that matters for the SSDs."

Note that we will keep the figure as it is in the manuscript as the above comment addresses the issue raised by the reviewer.

*Figure 3a: please indicate the pressure level for [U] either in the caption or on the figure itself*

We will indicate in caption that [u] is computed at 10 hPa and averaged between 45 and 75N.

*Is time smoothing applied for figure 4 and 5 and subsequent figures, or is figure 3 the only figure with time-smoothing? Please clarify either way. The reason I ask is that the text near line 258/259 seems to imply a time separation in the "synoptic" vs. planetary EPFD, however no such time separation appears in figure 4 and 5 (rather the planetary and "synoptic" waves change essentially simultaneously)*

The time smoothing is applied before plotting everywhere (no smoothing yields similar but slightly noiser results though). We will clarify this in the methods section (within paragraph on l. 244-247). We agree with the reviewer about the l. 258/9 - we will clarify that the events can occur simultaneously or synoptic waves slightly precede the planetary waves (case by case study reveals both options - not shown).

*Colorbar tick labels on figure 4,5, 6, etc.– please label the ticks symmetrically about zero.*

We have now made ticks symmetric about zero (figures not shown here; see Figs. 7-8 for an example).

*Figure 4 and 6: I found the equatorward/poleward tilt of EPF arrows to be somewhat confusing. I first thought this reflected some sort of propagation backwards in time, which of course makes no sense, but then I reread and understood. I don't think the question of equatorward vs. poleward propagation is particularly important to this study.*

While the meridional wave propagation is not emphasised in the manuscript, it generally still matters as the differences in wave propagation may lead to different responses of the atmosphere. We will clarify in the text and in the figure captions that arrows denote "*meridional* wave propagation, not propagation in time".

*Line 386- there is a positive stratospheric wind anomaly before the SSD in figure 6b, it is just weaker than in 6a,6e or 6e.*

The reviewer is correct - we will change the text accordingly.

*Lime 388 I don't understand this comment about a lack of preconditioning in figure 3a. Winds are clearly stronger than average before the SSD on figure 3a*

The reviewer is correct in that the winds are positive prior to SSDs in both datasets. However, in the model the wind anomaly remains below $0.5\sigma$ before an SSD, whereas in ERA-20C it is much stronger and exhibits a slight increase before an SSD. We will clarify these preconditioning differences in the text.

*Line 392- is this difference in [U] before the SSD between the "tropopause source" composite and the "surface source" composite actually statistically significant? I.e. a difference plot between panels b and e.*
[Figure]

We have plotted the difference between panels b and e (surface wave source cases minus tropopause wave source cases) - see Figs. 7,8 below. The difference is significant in ERA-20C (Fig. 8), but less so in the model (Fig. 7). We will discuss it this way and (as mentioned above) more carefully discuss the preconditioning differences where relevant. We will not include these figures in the manuscript.

**Technical comments**

*Line 121 in \*the\* lower*

We will add "the".

**References**

Plumb, R. Alan. "Planetary waves and the extratropical winter stratosphere." The Stratosphere, Dynamics, Transport and Chemistry, Geophys. Monogr 190 (2010): 23-41.

Matthewman, N.J. and Esler, J.G., 2011. Stratospheric sudden warmings as selftuning resonances. Part I: Vortex splitting events. Journal of the atmospheric sciences, 68(11), pp.2481-2504.

Held, I.M., Ting, M. and Wang, H., 2002. Northern winter stationary waves: Theory and modeling. Journal of climate, 15(16), pp.2125-2144.

Garfinkel, C.I., White, I., Gerber, E.P., Jucker, M. and Erez, M., 2020. The building blocks of Northern Hemisphere wintertime stationary waves. Journal of Climate,

33(13), pp.5611-5633.

Zakharov, V.E., L'vov, V.S. and Falkovich, G., 2012. Kolmogorov spectra of turbulence I: Wave turbulence. Springer Science & Business Media.

Kautz L-A, Polichtchouk I, Birner T, Garny H, Pinto JG. Enhanced extended-range predictability of the 2018 late-winter Eurasian cold spell due to the stratosphere. QJR Meteorol Soc. 2020;146:1040–1055. https://doi.org/10.1002/qj.3724

Hitchcock, P.: On the value of reanalyses prior to 1979 for dynamical studies of stratosphere–troposphere coupling, Atmos. Chem. Phys., 19, 2749–2764, https://doi.org/10.5194/acp-19-2749-2019, 2019.

Gerber, E. P. and Martineau, P.: Quantifying the variability of the annular modes: reanalysis uncertainty vs. sampling uncertainty, Atmos. Chem. Phys., 18, 17099–17117, https://doi.org/10.5194/acp-18-17099-2018, 2018.

**List of Figures with full captions**

Note that the template had a limit to the length of the caption, thus we provide full figure descriptions here, and figures are provided below with incomplete captions.

**Fig. 1**: Revised upscale cascade schematic from Fig. 2 of original manuscript.

**Fig. 2**: Composite analysis of downward impact in zonal mean zonal wind (latitudepressure vertical cross section) averaged between the lags 15 and 25 days (top row), 30 and 40 days (middle row), and 40 to 50 (bottom row) following (a,b,c) all SSW events, (d,e,f) SSWs not preceded by any wave source events, (g,h,i) SSWs preceded by tropopause wave source, and (j,k,l) SSWs preceded by surface wave source. The figure shows standardised zonal mean zonal wind anomalies (shading) and zonal mean zonal wind climatology (contours; contour interval is 5 m s$^{-1}$ with $0^{th}$ contour omitted for clarity, i.e. ...,-10, -5, 5, 10,...). Grey shading masks out data that are not significant at 95% level. Numbers in brackets denote number of events in each composite. Data are from the model.

**Fig. 3**: As in Fig. 2, but for ERA-20C data. Here composites are shown for zonal mean zonal wind (latitude-pressure vertical cross section) averaged between the lags 20 and 30 days (top row), 35 and 45 days (bottom row) following (a,b) all SSW events, (c,d) SSWs not preceded by any wave source events, (e,f) SSWs preceded by tropopause wave source, and (g,h) SSWs preceded by surface wave source.

**Fig. 4**: Map-composite analysis of downward impact in zonal wind at 950 hPa (U950) averaged between the lags -30 and 0 days (left column), and 0 and +30 days (right column) around SSW events: (top row) all SSWs, (second row) SSW events not preceded by a wave source, (third row) SSWs preceded by tropopause wave source, (fourth row) SSWs preceded by surface wave source. The bottom two rows show the same but for all wave source events around the central wave source date: (fifth row) all tropopause wave source events, (bottom row) all surface wave source events. The figure shows standardised U950 anomalies (shading) and U950 climatology (contours; contour interval is 4 m s$^{-1}$ with $0^{th}$ contour omitted for clarity, i.e. ...,-8, -4, 4, 8,...). Stippling denotes values that are significant at 95% level. Numbers in brackets denote number of events in each composite. Data are from the ERA-20C.

**Fig. 5**: As in Fig. 4, but for geopotential height at 500 hPa (Z500). Here the

[Figure]

figure shows standardised Z500 anomalies (shading) and Z500 climatology (contours; contour interval is 200 m, i.e. ...,4800, 5000, 5200,...).

**Fig. 6**: Similar to Fig. 3 from the original manuscript. Thick lines are as before, thin lines represent unsmoothed quantities.

**Fig. 7**: Difference between panels 6e and 6b (of the original manuscript). The plot shows standardised zonal mean zonal wind averaged between 40 and 60N for various lags. Data are from the model.

**Fig. 8**: Difference between panels 7e and 7b (of the original manuscript). The plot shows standardised zonal mean zonal wind averaged between 45 and 75N for various lags. Data are from ERA-20C.

a)

tropopause

nonlinear wave-
wave interaction

sink

pressure

k = 4   k = 2   k = 6

Jet stream

source

b)

pressure

k = 2   k = 2

Resonance/
diabatic/ non-
conservative

k = 2

k = 6

source

k = 2

k = 2

latitude

**Fig. 1.** Revised upscale cascade schematic from Fig. 2 of original manuscript.

[Figure]

**Fig. 2.** Composite analysis of downward impact in zonal mean zonal wind (latitude-pressure vertical cross section) averaged between the lags 15 and 25 days (top row), 30 and 40 days (middle row), and 40 to ...

[Figure]

**Fig. 3.** As in Fig. 2, but for ERA-20C data. Here composites are shown for zonal mean zonal wind (latitude-pressure vertical cross section) averaged between the lags 20 and 30 days (top row), 35 and 45 days...

all SSW; lag -30-0 (58)    all SSW; lag 0-30 (58)

SSW; no source; lag -30-0 (17)    SSW; no source; lag 0-30 (17)

SSW; tropopause s.; lag -30-0 (15)    SSW; tropopause s.; lag 0-30 (15)

SSW; surface s.; lag -30-0 (19)    SSW; surface s.; lag 0-30 (19)

all tropopause s.; lag -30-0 (291)    all tropopause s.; lag 0-30 (291)

all surface s.; lag -30-0 (292)    all surface s.; lag 0-30 (292)

**Fig. 4.** Map-composite analysis of downward impact in zonal wind at 950 hPa (U950) averaged between the lags -30 and 0 days (left column), and 0 and +30 days (right column) around SSW events: (top row) all ...

all SSW; lag -30-0 (58)   all SSW; lag 0-30 (58)

SSW; no source; lag -30-0 (17)   SSW; no source; lag 0-30 (17)

SSW; tropopause s.; lag -30-0 (15)   SSW; tropopause s.; lag 0-30 (15)

SSW; surface s.; lag -30-0 (19)   SSW; surface s.; lag 0-30 (19)

all tropopause s.; lag -30-0 (291)   all tropopause s.; lag 0-30 (291)

all surface s.; lag -30-0 (292)   all surface s.; lag 0-30 (292)

**Fig. 5.** As in Fig. 4, but for geopotential height at 500 hPa (Z500). Here the figure shows standardised Z500 anomalies (shading) and Z500 climatology (contours; contour interval is 200 m, i.e. ...,4800, ...

a) [u] about SSD events

b) EPdiv about tropopause wave source

c) EPdiv about lower tropospheric wave source

**Fig. 6.** Similar to Fig. 3 from the original manuscript. Thick lines are as before, thin lines represent unsmoothed quantities.

**[u] difference (surface-tropopause)**

**Fig. 7.** Difference between panels 6e and 6b (of the original manuscript). The plot shows standardised zonal mean zonal wind averaged between 40 and 60N for various lags. Data are from the model.

**[u] difference (surface-tropopause)**

**Fig. 8.** Difference between panels 7e and 7b (of the original manuscript). The plot shows standardised zonal mean zonal wind averaged between 45 and 75N for various lags. Data are from ERA-20C.

---

## Author Comment (AC2) · 5 Aug 2020

**Response to Reviewer 2**

We would like to thank the reviewer for carefully reading the manuscript, and for their detailed and constructive comments that will ultimately help improving the original manuscript. Below are our responses to the reviewer, which will be implemented in the revised manuscript in the next stage. Note that we have not provided exact manuscript corrections at this point, but we have provided the intended changes in detail; the line numbers/figure references in the reviewer's comments refer to the original manuscript. The reviewer's comments are in italics; our responses are in normal text.

[Figure]

*This paper addresses an important subject in stratosphere-troposphere coupling, which is how weak vortex events are ultimately forced. In particular, the authors concentrate on the appearance of a wave source around the tropopause in contrast to most work considering surface or mid-tropospheric wave sources. It is good to see new ideas explored, and I am looking forward to seeing this published. However, there are a few issues as described below which need to be addressed.*

**Major comments**

*1) My first major comment is about the model selection. a) The Held-Suarez/Polvani-Kushner model has been known to exhibit somewhat unrealistic dynamical behavior. Adding topography has mostly fixed the jet position issue, but what is probably more important in this particular study is the generally too low tropopause. I do not know what starting the stratospheric setup at 200hPa instead of 100hPa as described does to this bias, and there is no mention of any model validation in the manuscript. For instance, it is not true that dry dynamical cores do not produce SSWs with k=1 topography. The authors cite Sheshadri et al (2015), which is the model setup they use, but there are various other, improved versions of the dynamical core which do produce SSWs with wave-one (or realistic) topography. b) Throughout the manuscript, there are important differences between the model and ERA-20C, but those are not critically discussed. For instance, Fig. 4c) shows synchronous wave sources at both the surface and the tropopause, whereas 5c) indicates a possible propagation from the surface (at -10d) to the tropopause (around 0) and then back to the surface (at 5-10d and later). This is similar for Figs. 6a) and 7a). Fig. 6b) shows a surface source in the tropopause composite, whereas 7b) does not. Also, while shortly discussed in the manuscript, the preconditioning of the stratosphere in Figs 6 and 7 has a very different structure in the model vs ERA-20C, and so does the zonal mean zonal wind anomaly in Fig. 8. In Table*

*1, some percentages are close to ERA-20C, some are not, but there is no discussion of the confidence in the model results.*

*I think these differences are qualitative and require some discussion. Or the use of a different model with more realistic behavior.*

(a) The Polvani-Kushner setup has been extensively and successfully used for studying stratospheric dynamics and thus we feel comfortable using this particular setup - also because it offers comparisons to various previous studies. Also, as the main interest of this paper is the role of the tropopause wave source in coupling between troposphere and stratosphere (presently not well understood), a simplified mechanistic GCM is used, which allows further insight into the dynamical processes involved. The low tropopause issue is common in the tropics, but not in the midlatitudes - in fact, in our setup the tropopause is slightly higher up (thus the tropopause wave source is at higher altitude compared with ERA-20C). The 200 hPa transition was proposed as the tropopause layer in the Polvani-Kushner setup is too deep (i.e. unrealistic) and the 200 hPa transition somewhat helps with this. The displacement events are much harder to produce in the Polvani-Kushner setup than splits - even with k=1 topography as was clearly demonstrated by, e.g., Sheshadri et al (2015). If a more realistic setup or topography is used then also displacements become more common, but this is not the point of our study. We will further clarify these points in the manuscript.

(b) While we agree with the reviewer that these differences require more discussion, we would also like to point out that there are many similarities between the ERA-20C and the model (see also the response to Reviewer 1, comment 3). We will add further discussion about those differences in sections 4 and 5. Note that some similarities of the model with ERA-20C are remarkable and as such we are confident that the model results are representative of the zonal mean response, but less so for any local responses (e.g. Atlantic vs Pacific - again, see also the response to Reviewer 1, comment 3).

*2) Another important comment is a missing clear acknowledgment that most SSDs do not have any of the two wave sources. I understand the subjective selection to be able to study "clean" examples, and they do reveal interesting physics. But I feel that the manuscript is missing an estimate of how important the studied mechanisms really are. Another point is that only 10% of the tropopause source events are also SSDs (4% in the model), so what is happening with the other 90%? Same is true for surface source events.*

Note that in ERA-20C (an approximation to the real atmosphere) ∼30% of the SSDs are preceded by a tropopause wave source, thus the tropopause wave source is clearly important for the dynamics of SSDs in the real atmosphere. Moreover, this is a comparable percentage of SSDs that are preceded by the surface wave source, which have been studied extensively in the past. In the model the purpose is not to reproduce the same percentage of SSDs preceded by tropopause wave source, instead our aim is to study the events in a simplified setup. What is important here is that the model indeed exhibits tropopause wave source events and that these events can precede SSDs, allowing us to study the dynamics of wave source events and SSDs. The bias of the model towards smaller frequency of SSDs preceded by the tropopause wave source events may be a consequence of the model dynamics. Thus studying the reasons behind this bias could reveal some important aspects of the model dynamics, however this is beyond the scope of this study (left for future work).

Note also that the goal of our study is to advance our understanding of the role of the tropopause wave source in driving the SSDs, which are notoriously hard to predict. Thus, a better understanding of any precursor signal that can improve their predictability is useful, even if only a fraction of, e.g., wave source events result in SSDs (true for both surface and tropopause wave source events). That only 10% of the wave source events precede SSDs is a consequence of the stratospheric dynamics - i.e. some preconditioning in the stratosphere should be present for SSDs to occur (see, e.g., Hitchcock and Haynes 2016).

We will clarify these points in the revised manuscript.

**Minor comments**

*L 28-29: "Here note ..." should probably be moved to line 24, where Charney and Drazin (1961) is already discussed.*

We will move it to l. 25 after the sentence, instead of within it.

*L 31-32: This list of references is too long. It does not concentrate on the most important works but is not exhaustive either, and seems to mainly serve self-citation.*

We will cut some of the references.

*L 42: "suggest that an anomalous ..."*

We will change this phrase as the reviewer suggested.

*L 43: Same as L 31-32: too long but not exhaustive list of references. Also, shouldn't "Camara et al." read "de la Camara et al."?*

We thank the reviewer for pointing out the "de la Camara" issue (official bibtex files had it wrong). As for the reference list - we will remove some of them.

*L 102: "Note that the wave decay..." is not necessary as this is discussed immediately afterward.*

We will remove the sentence.

*L 113-14: "Therefore, even...": you just concluded that div(F) = 0, so where is the "significant increase"? I am sure you are trying to say something different, but this*

*is confusing as written.*

If there is a significant decrease and then a significant increase (exceeding $0.75\sigma$), and they equal each other, integral over time gives us div(F)=0. We will rephrase the sentence as: "Therefore, even if the EP flux divergence exceeds a set threshold and appears as though there is a wave source, this is merely representing a decay of a wave, and thus we will refer to it as an apparent wave source."

*L 124: "(sink)": this is a bit confusing, as at first I thought this was meant to mean that the waves are sinking downward. Maybe move this into the parenthesis "(negative EP flux divergence, i.e. a sink)" or similar.*

We will move "sink" after "negative EP flux divergence" within parantheses as suggested by the reviewer.

*L 131: Note that the newly created k=2 wave can also cancel an existing k=2 wave if the phase is opposite. Try and be more careful when describing the triad interactions.*

We agree with the reviewer and will therefore mention the cancellation as well.

*Paragraph 2.2: Note that while valid, the triad interactions can (and probably will) only convert waves partially, i.e. there is going to be partial dissipation and resonance, plus other effects, so a comment on the testability of this process would be welcome.*

We agree with the reviewer. Reviewer 1 has also pointed out some caveats about the 'resonance' we were mentioning. Resonance was never meant as the only possible mechanism here, thus we will add further options for enhanced EPFD, and more carefully discuss the resonance part in the majority of the manuscript (see also the response to Reviewer 1). We will add a comment about the dissipation in the revised

manuscript.

*Paragraph 3.1: Maybe change title to "Model & Data"*

We will change the title as the reviewer suggests.

*L 164: 0 hPa is at infinity, so surely the model top is somewhere else?*

The top model interface is at 0 Pa (which is at the top of the atmosphere in pressure coordinates, not necessarily in the infinity), but the top half-level is at $\sim 7$ Pa. See also Held and Suarez (1994) - section 3a, where they say that the model top is formally at 0 Pa.

*L 193: 20-days $->$ 20 days*

We will change it accordingly.

*Paragraph 3.3: It is ok to subjectively select a few "clean" events, but there should be an estimate of the relative importance of what you are filtering out, i.e. you select a few events which show your mechanism, but in the grand scheme of things, how important is that mechanism?*

We mentioned that generally speaking the two are hard to separate as they tend to occur simultaneously, likely amplifying each other. Thus, the two mechanisms are important - we can also see them in an average over all events (as mentioned in the text). However, other accompanying effects cannot be excluded, i.e. rare events may have a different origin. We will clarify this further.

*L 332: remove "(section 5)" as that's the very next thing.*

We will remove this remark.

*L 363-365: This relates to some of the comments above: Do you have any interpretation as to why only 10/4% result in SSDs and what happens with the remaining 90/96%?*

I have mentioned this above as well - this is likely because the stratosphere has to be in the right state (e.g. Hitchcock and Haynes 2016). We will add this comment in.

*L 424: "into deep stratosphere": maybe change to "deep into the stratosphere"*

We will change it.

*L 439-440: Do the authors have any suggestion about how one could check for selftuning or downscale cascade? Not necessary, but would be very helpful for future work.*

For downscale cascade we could use a similar approach as used here for upscale cascade, however the self-tuning resonance is a more complex issue; both are well beyond the scope of this study. This comment was merely meant as a remark - we will mention that this could be looked into in the future.

*L 444: Again, there are others with this idea*

We will change the list to include other authors as well.

*L 491-492: The difference between model and ERA-20C seems much larger than the difference between tropopause and source wave source events. Can you then really assert that there is a difference between tropopause and surface wave source in terms of downward impact?*

Note that we made a mistake when producing Fig. 8, which will now be updated. The revised figures (see Figs. 2-3 in response to reviewer 1) that will replace Fig. 8 of the original manuscript show more similarities between the model and ERA-20C

than the original Fig. 8. While some differences remain, the differences between the tropopause and surface wave source are generally larger than differences between the model and ERA-20C. The difference between the surface and tropopause wave source is especially pronounced in ERA-20C (see also response to reviewer 1, comment 3).

*L 515-516: "While cases..." there is something missing in this sentence.*

This will be changed to: "While there are cases..."

*L 517: occurs, the*

We will add the comma as the reviewer suggested.

––––––––––––––––––––––––––

---

## Author Response (AR2)

**Response to Reviewer 1**

We would like to thank the reviewer for carefully reading the manuscript, and for their detailed and constructive comments that have helped improving the original manuscript. Below are our responses to the reviewer, where all figures that were produced in response to the reviewer's comments are at the end of the responses to the reviewer 1; the line numbers/figure references in the reviewer's comments refer to the original manuscript; the line numbers in the responses refer to the manuscript with tracked changes (denoted in bold). The reviewer's comments are in italics; our responses are in normal text.

*The authors have satisfactorily addressed my major comments. Here are some final minor changes before publication:*

*1. Line 33 and 35: the authors imply that a 1.95std deviation anomaly of tropospheric wave flux is not "strong", and a SSD or SSW preceded by such an anomalous tropospheric wave flux event is not associated with a tropospheric precursor. I would beg to differ to both of these implications. To be constructive, the simplest way to revise this is to replace "strong" with "extreme" on lines 33 and 35.*
We have changed it as the reviewer suggested. See l. 33, 35.

*2. Line 71: this sentence reads a bit funny, suggest rewriting*
We have rewritten the beginning of the sentence. See l. 71.

*3. Line 76-77 I had trouble following the logic of this sentence. That being said, I accept the authors rebuttal to my original criticism regarding using QG theory and turbulence, and the revised section 2 is much better than before.*
We meant to say that coupling of the waves and the mean flow is nonlinear (even in a simple QG framework). We have rephrased the sentence and emphasised 'coupling'. See l. 76-77.

*4. The paragraph from line 128 to line 148 is rather long, and would benefit from being split in two or even three (e.g. at line 142)*
We have split the paragraph as suggested. See l. 142-143.

*5. Line 303 by \*an\* equally strong*
*6. line 385 prior to \*the\* wave source event*
We have added "an" and "the" as suggested. See l. 305, 387.

*7. Is the difference in the tropospheric impact between tropopause wave source SSWs and surface wave source SSWs statistically significant? Maybe add a difference plot? The difference plot should be created for both quasi-reanalysis and the model, and it is conceivable that the limited sample size leads to a lack of significance. If the difference indeed isn't significant, then the discussion in section 5 (as well as the conclusions section and abstract) should be modified accordingly.*
After computing the difference and its significance as the reviewer suggested, we found some regions that are statistically significant (Figs. 1, 2 below). The significant differences are limited to the positive values - e.g., 30-40N in Fig. 1 top row, 50-60N in Fig. 1 second row; or 60-70N in Fig. 2 top row. In the model this reflects the more significant downward impact following the tropopause wave source events in the top row (Fig. 1 here and Fig. 8 in the manuscript), and the more significant downward impact following the surface wave source events in the second row (Fig. 1 here and Fig. 8 in the manuscript); whereas in ERA-20C it reflects the different downward impacts of the two wave source events (top row in Fig. 2 here and in Fig. 9 in the manuscript).

Since the significant regions are small we have modified the text to reflect this (e.g., omitting 'robust', adding 'indicated'/'suggested'), though we have not provided these additional figures in the main manuscript (for brevity). See bold text in the abstract, section 5.2.3, and conclusions.

[Figure]

**Figure 1.** Difference between downward impact in (standardised) zonal mean zonal wind following SSWs preceded by the tropopause and surface wave source (shading). The top row is for lags 15-25 days, the second row for lags 30-40 days, and the bottom row is for lags 40-50 days following the SSWs. Contours represent climatological zonal mean zonal wind (contour interval is 5 m/s with $0^{th}$ contour omitted for clarity), and grey shading masks out data that are not significant at 95% level (via two-tailed t-test's comparison of the means). Data are from the model.

[Figure]

**Figure 2.** As in Fig. 1, but for ERA-20C. Here the top row is for lags 20-30 days, and the bottom row is for lags 35-45 days following the SSWs.

**Response to Reviewer 2**

We would like to thank the reviewer for carefully reading the manuscript, and for their detailed and constructive comments that that have helped improving the original manuscript. Below are our responses to the reviewer, where the line numbers/figure references in the reviewer's comments refer to the original manuscript; the line numbers in the responses refer to the manuscript with tracked changes (denoted in bold). The reviewer's comments are in italics; our responses are in normal text.

*I find the manuscript improved with just a few minor issues remaining. They are:*

*L134-139: Turbulence theory: I realise this text has been changed in response to other referee comments, but I believe there is still a certain lack of accuracy in the statement as written now. Triad interactions in 2D turbulence are explicitly defined as the interactions of three waves with wavenumbers k1 = k2+k3. So they are wave- wave interactions of a finite number of distinct waves (three of them). And if k1,k2,k3 are small numbers, they are wave-wave interactions of a finite number of distinct large waves. So the statement in the manuscript saying the authors concentrate on these interactions "rather than [...] 2-D turbulence)" is not accurate. Please clarify.*

We are considering a finite number of triads, not the number of waves within a triad (three waves interact in a triad by definition). Thus, this is not a continuous 2D turbulence - there is a cut-off - instead we consider a finite number of triad interactions that only occur within the large scale wave spectrum (i.e., a subset of the triad interactions involved in the full 2D turbulence). We have modified the text to further clarify this - see l. 137-139.

*L 216: "SSWs are defined as a subset of the identified SSDs". How do these SSWs compare to the classical Charlton & Polvani (2007) SSWs? Are all SSDs which also reverse zonal flow CP07 SSWs? What about the other way round? It would be good to have a statement here to get a feeling of how similar the here discussed SSWs are to the CP07 SSWs.*

Table 1 in the supplement of Birner and Albers (2017) provides a comparison of the SSD and SSW (CP07) dates in ERA-Interim. This shows that not all SSWs (CP07) are SSDs and not all SSDs are SSWs. However, in our model the reviewer's suggestion (SSDs with SSWs are also CP07 SSWs) is true, but not in ERA-20C or other reanalyses. Note that we are not implying that some SSWs do not exhibit abrupt decelerations, they may simply be slower, weaker or not occurring within the specified time-frame (i.e., are inconsistent with our definitions)! Examples of this can be secondary SSWs or polar night jet oscillations, where typically the first SSW may be related to strong decelerations but the subsequent ones might not.

Note that composites over SSWs associated with SSDs and all SSWs as per CP07 yield qualitatively similar results (see Figs. 1, 2). Thus, even though the two indices are generally not the same, composites do not differ much. Since the focus here is the dynamical evolution around the strong deceleration events, SSDs, we have used the SSW definition described in the manuscript (for consistency) instead of the CP07 definition.

We have now explicitly mentioned these points. See l. 219-221.

*L 224-225: This change after my previous comments is not satisfying, as it still suggests there are no models were k=1 forcing does produce SSWs. So, why not make this statement much simpler (and more accurate), by saying ".., however our model does not exhibit SSWs when forced with k=1 forcing (Table 1 in Sheshadri et al., 2015)."*

We have changed it as suggested. See l. 227-228.

*L229-270: I got very confused about 10-day running means here. You say you first smooth the data with a 10-day running mean (L 229), then you apply a 10-day smoothing before compositing (L 251), and then you smooth the data with a 10-day running mean before plotting (L 270). How many times, exactly, do you apply a 10- day running mean until you get to the final, plotted results? If it's more than once, why is this necessary?*

We apologise for the confusion. The smoothing/running mean was done only once for each diagnostic and/or plot. The additional clarifications were added as per reviewer 1's suggestions in the previous round of reviews - we merely clarified that data was smoothed before plotting (l. 251, l. 270). We have changed the text (e.g., adding "as mentioned above") to avoid confusion on the number of times we smoothed the data (see l. 253, 270-271). Generally we smoothed the data to get the index, then composited smoothed data over the index.

50   *L522, 523: Instead of just referring to Fig 6b,e and 7b,e, it would be much clearer if you could discuss the physical quantity and add the figure information to the discussion, maybe in parentheses.*

   We have changed the text as suggested. See l. 524-525.

[Figure]

**Figure 1.** Composite analysis of downward impact in zonal mean zonal wind (latitude-pressure vertical cross section) averaged between the lags 15 and 25 days (top row), 30 and 40 days (middle row), and 40 to 50 (bottom row) following (a1,b1,c1) all SSW events identified via $[u]$ at 10 hPa and 60°N (CP07), and (a,b,c) all SSW events preceded by an SSD. The figure shows standardised zonal mean zonal wind anomalies (shading) and zonal mean zonal wind climatology (contours; contour interval is 5 m s$^{-1}$ with 0$^{th}$ contour omitted for clarity, i.e. ...,-10, -5, 5, 10,...). Grey shading masks out data that are not significant at 95% level. Numbers in brackets denote number of events in each composite. Data are from the model.

[Figure]

**Figure 2.** As in Fig. 1, but for ERA-20C data. Here composites are shown for zonal mean zonal wind (latitude-pressure vertical cross section) averaged between the lags 20 and 30 days (top row), 35 and 45 days (bottom row) following (a1,b1) all SSW events identified via $[u]$ at 10 hPa and 60°N (CP07), and (a,b) all SSW events preceded by an SSD.

**Tropopause-level planetary wave source and its role in two-way troposphere-stratosphere coupling**

Lina Boljka[1] and Thomas Birner[2]

[1]Department of Atmospheric Science, Colorado State University, Fort Collins, Colorado, USA
[2]Meteorological Institute, Ludwig-Maximilians-Universität München, Munich, Germany

**Correspondence:** Lina Boljka (lina.boljka@colostate.edu)

**Abstract.**

Atmospheric planetary waves play a fundamental role in driving stratospheric dynamics, including sudden stratospheric warming (SSW) events. It is well established that the bulk of the planetary wave activity originates near the surface. However, recent studies have pointed to a planetary wave source near the tropopause that may play an important role in the development of SSWs. Here we analyse the dynamical origin of this wave source and its impact on stratosphere-troposphere coupling, using an idealised model and a quasi-reanalysis. It is shown that the tropopause-level planetary wave source is associated with nonlinear wave-wave interactions, but it can also manifest as an apparent wave source due to transient wave decay. The resulting planetary waves may then propagate deep into the stratosphere, where they dissipate and may help to force SSWs. **Our results indicate that SSWs preceded by both the tropopause and the surface wave source events tend to be followed by a weakened tropospheric zonal flow several weeks later. However, while in the case of a preceding surface wave source event this downward impact is found mainly poleward of 60°N, it appears to be the strongest between 40-60°N for SSWs preceded by tropopause wave source events. This suggests that tropopause wave source events could potentially serve as an additional predictor 
[revised manuscript text omitted]

**The different** signal in tropospheric zonal flow following SSWs preceded by surface versus tropopause wave source events in ERA-20C and in the model as well as the presence of two different types of wave source events, also suggest that care must be taken when using indices, such as 100 hPa heat flux (see also de la Cámara et al., 2017). This is because: (i) the waves occurring at 100 hPa can be excited at the surface or at the tropopause (shown here), or even internally within the stratosphere (e.g., Plumb, 1981); and (ii) the SSWs preceded by the tropopause wave source **indicate** a downward impact (zonal flow deceleration) in 40-60°N latitudinal band, whereas SSWs preceded by surface wave source **indicate** an opposite surface signal with zonal flow deceleration further poleward.

Furthermore, we have also shown that the polar vortex preconditioning, i.e., strengthening of the polar vortex prior to SSD events (which can also be used as one of the precursors to SSDs; e.g., Jucker and Reichler, 2018) could be different between the SSDs preceded by tropopause and surface wave source events. This means that SSDs preceded by the tropopause wave source events are generally related to weaker zonal mean zonal wind strengthening (i.e., negative PV gradients) prior to SSDs (weaker polar vortex preconditioning is necessary), whereas SSDs preceded by surface wave source events are related to significant strengthening of the zonal flow prior to SSDs (i.e., stronger polar vortex preconditioning is necessary).

[revised manuscript text omitted]

Domeisen, D. I. V., Sun, L., and Chen, G.: The role of synoptic eddies in the tropospheric response to stratospheric variability, Geophysical Research Letters, 40, 4933–4937, https://doi.org/10.1002/grl.50943, 2013.

[revised manuscript text omitted]